# Separate luminous structures leading positive leader steps

Shengxin Huang[1,6], Weijiang Chen [2✉], Zhong Fu[3], Yufei Fu[4], Nianwen Xiang[4], Xinjie Qiu[3], Weidong Shi[1], Dengfeng Cheng[3] & Zhiyuan Zhang[5]

The physics governing the propagation of lightning leaders and long spark leaders is still not well understood. Positive and negative leaders seem to behave differently. Negative leaders develop in a step manner, guided by the separate luminous structures termed space stems and space leaders. Positive leaders, on the other hand, are generally thought to have no separate luminous structure involved in their propagation. However, a separate luminous structure observed in a positive leader discharge had been reported in recent literature, suggesting that positive leaders may similarly do steps to negative leaders under certain conditions. Here we report the observation of the positive leader step led by a separate luminous structure at high humidity in laboratory lightning-like discharges. We also found the streamer-like common zone connecting the primary leader channel with the separate luminous structure, as well as the bi-directional development of the separate luminous structure. We hope that these findings would contribute to a better understanding of the nature underlying positive long spark leaders and lightning leaders.

[1] High Voltage Department, China Electric Power Research Institute, Beijing 100192, China. [2] State Grid Corporation of China, Beijing 100031, China. [3] Electric Power Research Institute, State Grid Anhui Electric Power Company, Hefei 230022, China. [4] School of Electrical Engineering and Automation, Hefei University of Technology, Hefei 230009, China. [5] School of Electrical and Electronic Engineering, Huazhong University of Science and Technology, Wuhan 430074, China. [6] Present address: School of Electrical Engineering and Automation, Hefei University of Technology, Hefei 230009, China. ✉email: chenweijiang.China@Gmail.com

Lightning is a commonly experienced yet still mysterious natural phenomenon. The lightning's physical nature is that the thermally ionized plasma channels, termed positive and negative leaders, extend in virgin air[1]. The polarity asymmetry of positive and negative leaders, which describes the difference in macroscopic behavior between positive and negative leaders, is a consensus broadly accepted by lightning physicists[2]. Negative leaders propagate in discrete steps, led by the separate luminous structures called space stems and space leaders, with rapid elongations and sharp channel illuminations[3–10]. Positive leaders usually propagate in a quasi-continuous fashion, with optically unresolvable steps whose length is comparable to the leader tip size and continuous current[11,12]. During the step, negative leaders emit copious very high frequency (VHF) radiation that can be remotely sensed and imaged[8–10,13–17], while the positive leader tips are often VHF radiation silent[9,18–20]. The VHF radiation emitted from positive leader bodies is attributed to the needle-like discharge recently discovered[20–22] and recoil leaders[18,23].

Under particular conditions, positive leaders could also exhibit steps/restrikes, which can cause rapid elongations and are associated with large current pulses and sharp channel reilluminations, during the quasi-continuous development[24–29]. Without realizing the existence and the role of space leaders in positive leader discharges, previous studies inferred that air humidity has some action enhancing the instabilities in the leader quasi-continuous advancement[24], explaining the occurrence of positive leader steps at high humidity. Recently, a separate luminous structure had been observed inside the leader corona zone in positive leader discharges[30]. The interesting result highlights a critical question of whether positive leaders could do step into a

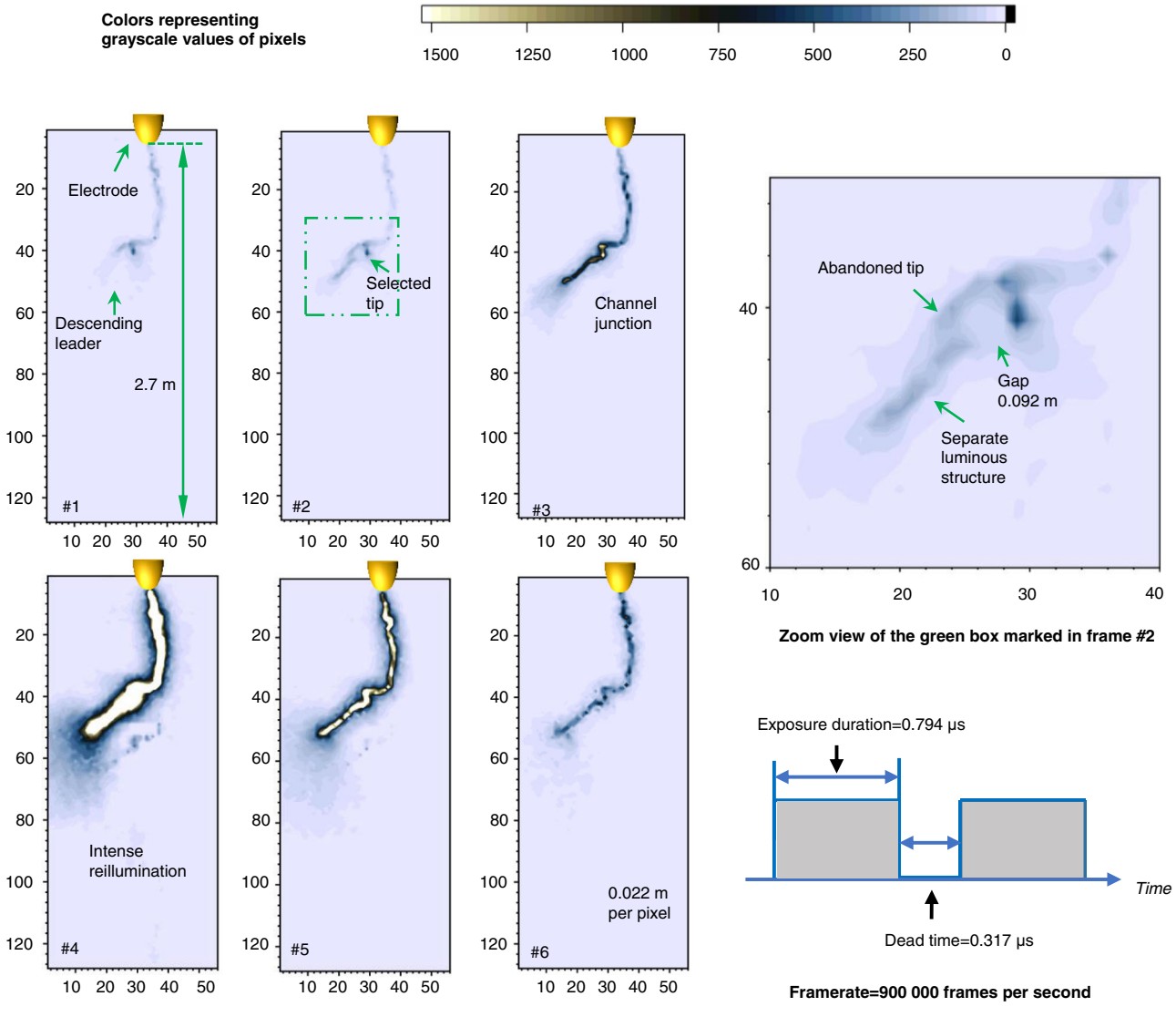

**Fig. 1 Frames recording a positive leader step led by a separate luminous structure.** The original high-speed video frames recording the positive leader step are grayscale images with 12-bit pixel depth, photography by a Photron FASTCAM SA-Z high-speed camera. A Canon TV-16 lens (25 mm 1:0.78) was used. The framerate was set as 900,000 frames per second. The resolution of these frames is 56 (in width) × 128 (in height) pixels. The field of view covered a part of the air gap (2.7 m). To enhance visibility, these frames were pseudo-colored and shown in the figure. The color of pixels in these frames represents the value in the grayscale matrix extracted from the original frames. Pixels with grayscale exceeding 1500 are displayed in white. The coordinate marked in these frames represents the row and column numbers of the grayscale matrix. The 2.7-m long gap occupies 123 rows in the grayscale rows, indicating the spatial resolution of these frames is about 0.022 m per pixel. During the experiment, the relative humidity is 78 ± 2%, the temperature is 301 ± 0.21 K, the air pressure is 1 atm, and the calculated absolute humidity is 21.20 ± 0.3 g/m³.

similar mechanism as negative leaders. Unfortunately, it was not clear if the separate luminous structure resulted in a step or not in the observation[30]. The nature of the steps/restrikes occurring during the quasi-continuous positive leader development at high humidity is still not well understood.

Here we report the positive leader steps led by separate luminous structures recorded by high-speed cameras. In addition, two critical characteristics of the separate luminous structure that leads positive leader steps are also presented. One is the streamer-like common zone that connects the main leader channel with the separate luminous structure, and the other is the separate luminous structure's bi-directional development. Our result implies that, at least at high absolute humidity, positive leaders can form steps due to the merging of a separate luminous structure and the primary leader channel, similar to the steps of negative leaders, which are formed due to merging the space leader with the main negative leader.

## Results

**The positive leader step led by a separate luminous structure.** A sequence of consecutive frames in Fig. 1 depicts a positive leader step guided by a separate luminous structure in a positive long spark discharge. The positive long spark discharge was produced in a rod-to-plane air gap with a length of 4 m on August 20, 2020.

In frame #2, we see the separate luminous structure that emerged ahead of the propagating leader tip. The propagating leader tip was branched, as illustrated in frames #1 and #2. In frame #2, a gap is obvious between the upper end of the separate luminous structure and the selected leader tip. The gap is about 0.092 m in length. By comparing the morphology of the leader channel recorded by frame #3 to that of the separate luminous structure, it is indicated that the separate luminous structure became the newly added portion of the leader channel during the exposure duration of frame #3, causing the abrupt elongation of the descending leader channel. The intense illumination of the merged leader channel is captured in the subsequent frame #4. Notably, due to the nearly straight shape of the separate luminous structure, the extension path of the positive leader is roughly straight during the step, as reported in a previous study[30], which is fundamentally different from the tortuous channel development trajectory observed during quasi-continuous propagation of positive leaders.

It is essential to highlight that the gap between the separate luminous structure and the abandoned tip of the branched leader is not clear. Assuming that the separate luminous structure is a port of the leader channel quasi-continuously extending from the abandoned tip, this leader segment is charged positively. The selected tip of the branched leader would move away from this leader segment, due to the Coulomb force. The channel junction shown in frame #3 would not occur. As a result, it could be deduced from physics that the separate luminous structure is not actually connected to the abandoned tip. The insignificance of the gap between the separate luminous structure and the abandoned tip may be attributable to the optical illusion caused by two-dimensional photography.

Figure 2 illustrates the discharge current of the positive leader discharge event. According to the synchronization of the exposure time of the frames shown in Fig. 1 with the discharge current, the current pulse associated with the step was registered at approximately $t = 69\,\mu s$. The instant is within the exposure duration of frame #2 in Fig. 1 that depicts the separate luminous structure. The current pulse is superimposed on a continuous and stable current, whose value is about 0.4 A. The current peak is about 13.2 A. By and large, the current pulse exhibits a sharp increase, similar to the step current waveform presented in the

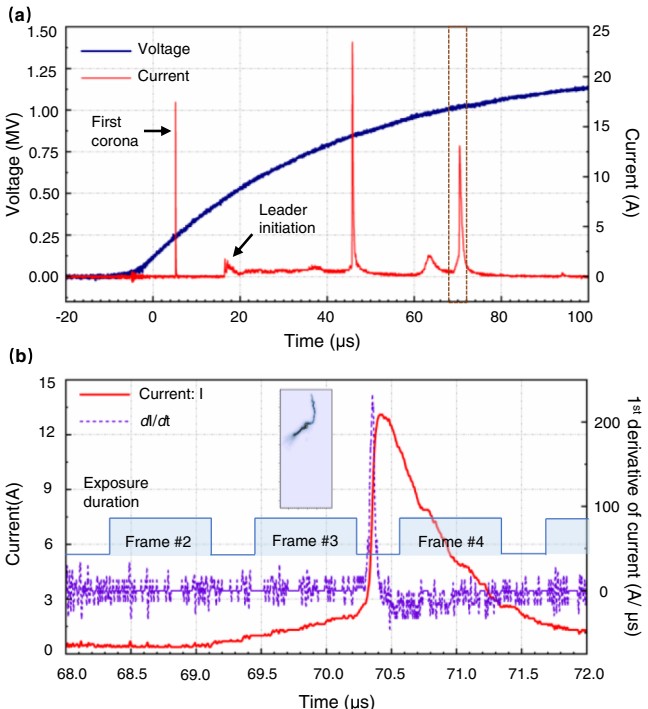

**Fig. 2 Discharge current associated with the step led by a separate luminous structure.** Panel **a** is the discharge current of the positive leader step, synchronized with the applied impulse voltage waveform. Panel **b** shows the detail of the current in the brown box marked in the above panel. The current is synchronized with the exposure strobe of the high-speed camera. The first derivative of the current was calculated, shown in the figure to indicate the variation of the current rise rate.

literature[30], where the existence of a separate luminous structure in the positive leader discharge was first reported. The discharge current rose from 10% of the current peak (1.32 A) to 90% of the current peak (11.88 A) in 556 nanoseconds. From a more detailed perspective, two stages can be distinguished based on the current waveform. From $t = 69\,\mu s$ to $t = 70.3\,\mu s$, the discharge current rose slowly at a low rate of rising. The stage lasted approximately 1.3 microseconds. The current increased from 0.4 to 2.78 A during the stage. Since $t = 70.3\,\mu s$, the discharge current rose sharply from 2.78 A, reaching a peak at $t = 70.41\,\mu s$. The stage lasted about 110 nanoseconds. The step current pulse's steep front is attributable mostly to the current's sharp increase in the latter stage.

As illustrated in Fig. 2, the exposure duration for frame #3 is included in the stage during which the current rose slowly. The stage during which the current increased sharply began following the end of the frame's exposure duration. A merged leader channel is apparent in the frame, implying that the connection between the separate luminous structure and the main leader channel had been completed prior to the end of the exposure duration for the frame. This result implies that the current began to rise sharply after the separate luminous structure had been merged with the main leader channel.

**Features of the separate luminous structure leading positive leader steps.** In a positive leader discharge produced in a rod-to-plane air gap with a 5 m length on September 9, 2021, a streamer-like common zone connecting the primary leader channel with the separate luminous structure was observed. Morphologically, the streamer-like common zone is a dense luminous spindle.

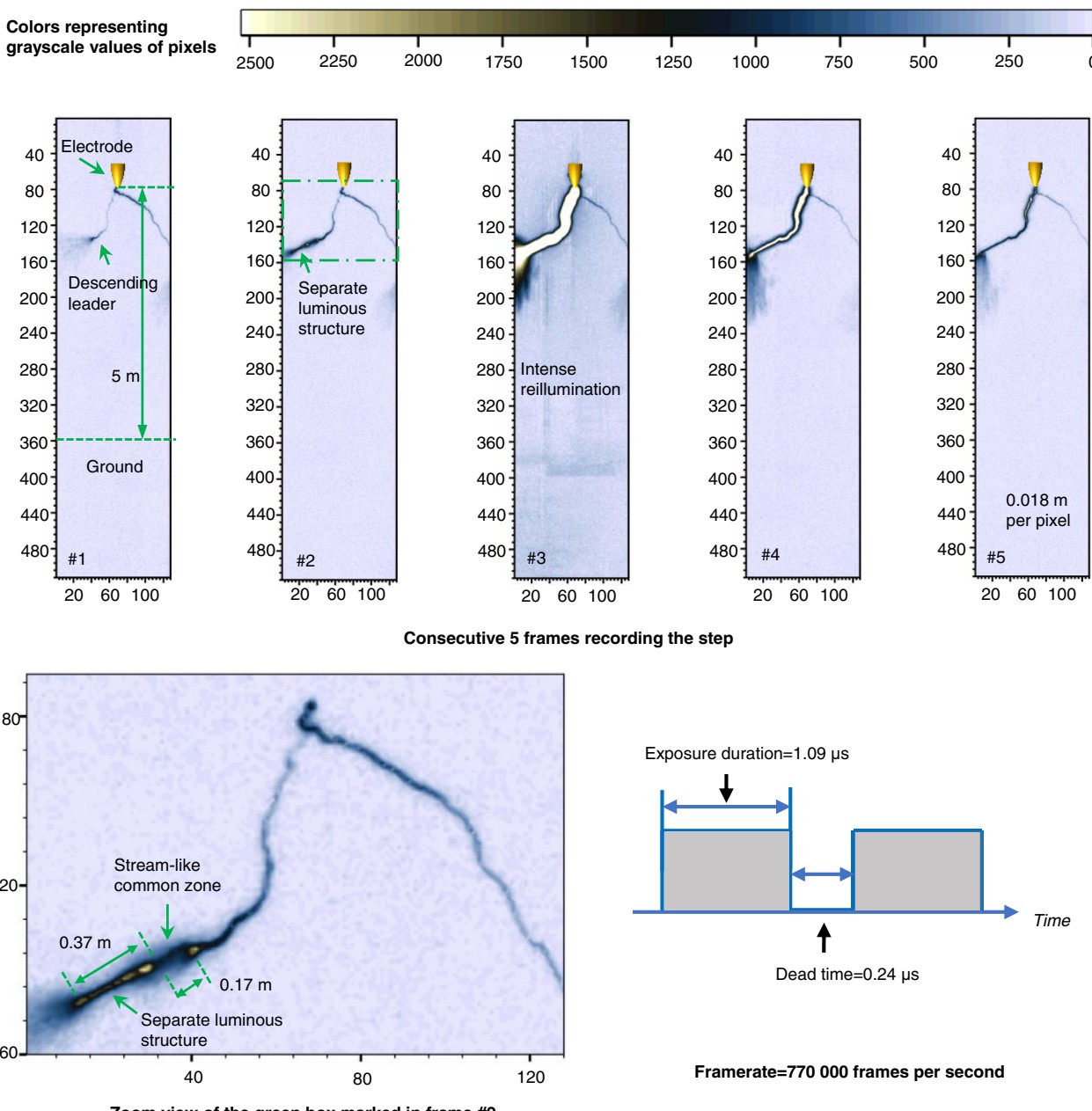

**Fig. 3 The streamer-like common zone connecting the separate luminous structure with the main channel.** The original high-speed video frames recording the positive leader step are grayscale images with 12-bit pixel depth, photography by a Phantom TMX 7510 high-speed camera. A Nikon lens (50 mm 1:1.2) was used. The framerate was set as 770,000 frames per second. The exposure duration of each frame is 1.09 μs, and the interval between two adjacent frames (dead time) is 0.24 μs. The resolution of these frames is 128 (in width) × 512 (in height) pixels. The field of view covered the whole air gap (larger than 5 m). To enhance visibility, these frames were pseudo-colored and shown in the figure. The color of pixels in these frames represents the value in the grayscale matrix extracted from the original frames. Pixels with grayscale exceeding 2500 are displayed in white. The coordinate marked in these frames represents the row and column numbers of the grayscale matrix. The 5-m long gap occupies 278 rows in the grayscale rows, indicating the spatial resolution of these frames is about 0.018 m per pixel. During the experiment, the relative humidity is 75 ± 2%, the temperature is 300.5 ± 0.21 K, the air pressure is 1 atm, and the calculated absolute humidity is 19.83 ± 0.77 g/m³.

The separate luminous structure then led a positive leader step. A sequence of consecutive frames in Fig. 3 depicts the discharge before, during, and after the positive leader step.

Frames #1–#3 in Fig. 3 record the quasi-continuous development of the descending positive leader. The emergence of a separate luminous structure later leading a positive leader step was imaged by the following frame #4. The separate luminous structure is a long and straight segment and appears ahead of the descending leader tip. The length of the separate luminous structure is about 0.37 m. A streamer-like common zone, morphologically similar to the common streamer zone in the breakthrough phase[31], connects the upper end of the separate luminous structure to the primary leader channel tip. The streamer-like common zone is around 0.17 m in length. The next frame shows that the primary leader channel abruptly extended and intensely illuminated in its exposure duration, indicating that a positive leader step occurred. The newly added segment is also nearly a straight line. Comparing the morphology of the leader

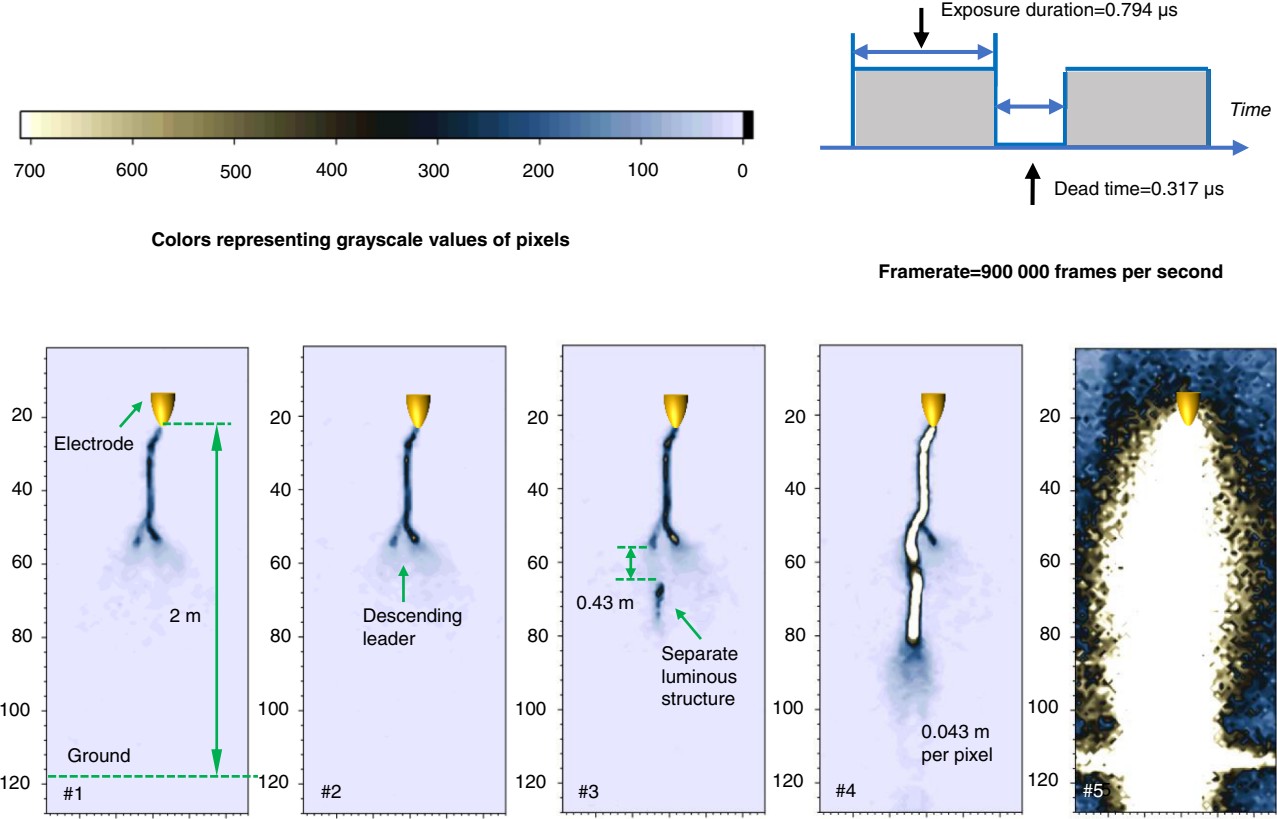

**Fig. 4 Bi-directional development of the separate luminous structure.** The original high-speed video frames recording the positive leader step are grayscale images with 12-bit pixel depth, photography by a Photron FASTCAM SA-Z high-speed camera. A Canon TV-16 lens (25 mm 1:0.78) was used. The framerate was set as 900,000 frames per second. The exposure duration of each frame is 0.794 µs, and the interval between two adjacent frames (dead time) is 0.317 µs. The resolution of these frames is 56 (in width) × 128 pixels (in height). The field of view covered the whole air gap (larger than 4 m). To enhance visibility, these frames were pseudo-colored and shown in the figure. The color of pixels in these frames represents the value in the grayscale matrix extracted from the original frames. Pixels with grayscale exceeding 700 are displayed in white. The coordinate marked in these frames represents the row and column numbers of the grayscale matrix. The 4-m long gap occupies 92 rows in the grayscale rows, indicating the spatial resolution of these frames is about 0.043 m per pixel. During the experiment, the relative humidity is 74 ± 2%, the temperature is 301 ± 0.21 K, the air pressure is 1 atm, and the calculated absolute humidity is 20.11 ± 0.77 g/m$^3$.

channel recorded by frame #5 and the separate luminous structure, it is suggested that the separate luminous structure became the newly added section of the leader channel in the step, causing the abrupt elongation of the descending leader channel.

In another positive leader discharge produced in a rod-to-plane air gap with a 4 m length on June 22, 2020, the sequence of consecutive frames recording the bi-directional development of a separate luminous structure was obtained and is shown in Fig. 4.

Frames #1 and #2 record the quasi-continuous development of the positive leader. Comparison of frame #3 and frame #4 shows the bi-directional development of the separate luminous structure. A separate luminous segment was first imaged by frame #3. The separate luminous segment emerged ahead of the leader channel tip and at about 0.43 m from the descending positive leader tip. The brightness of the floating luminous formation is close to that of the primary leader channel. As shown in the next frame, the separate luminous structure extended and almost contacted the main leader channel tip. The length of the separate luminous structure shown in frame #4 is about 1 m. It is reasonable to estimate that the merging of the separate luminous structure with the main channel extended the leader channel by about 1 m. This length is close to the length of the steps of the positive leader previously measured[30]. The bi-directional

development of a separate luminous structure was recorded only in two consecutive frames (frames #3 and #4), indicating that the separate luminous structure emerged after the end of frame #2's exposure duration but before the end of frame #3's exposure duration, and developed to a length of about 1 m at the end of frame #4's exposure duration.

During the bi-directional development of the separate luminous structure, the upper end of the separate luminous structure propagated backward (toward the descending leader tip), and the lower end of the separate luminous structure developed forward. The initiation point of the separate luminous structure emitted light for a longer period of time compared to the other positions of the separate luminous structure during its growth, so it is reasonable to assume that the pixel point with the largest grayscale value on the separate luminous structure shown in frame #3 is its initiation point. Considering the framerate of the high-speed camera, it is could be estimated that the average forward development velocity of the separate luminous structure is less than $6.53 \times 10^5$ m/s and greater than $3.27 \times 10^5$ m/s, and the average backward development velocity of the separate luminous structure is less than $2.47 \times 10^5$ m/s and greater than $1.24 \times 10^5$ m/s. The average forward development velocity of the separate luminous structure was obviously greater than the

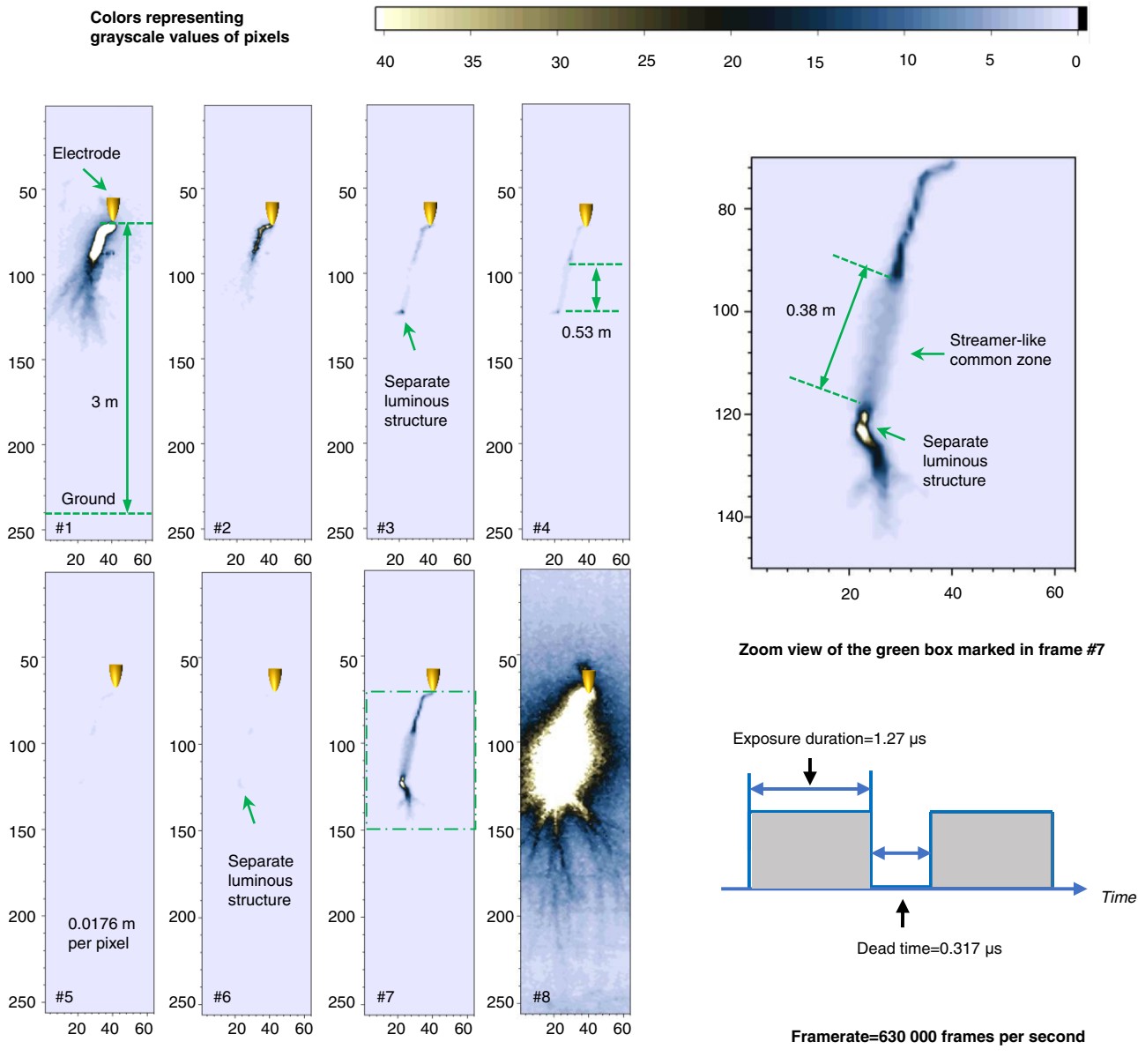

**Fig. 5 Frames recording a typical negative leader step led by a separate luminous structure.** The original high-speed video frames recording the negative leader step are grayscale images with 8-bit pixel depth, photography by a Photron FASTCAM SA-Z high-speed camera. A Canon TV-16 lens (25 mm 1:0.78) was used. The framerate was set as 630,000 frames per second. The exposure duration of each frame is 1.27 µs, and the interval between two adjacent frames (dead time) is 0.317 µs. The resolution of these frames is 64 (in width) × 256 (in height) pixels. The field of view was larger than 3 m. To enhance visibility, these frames were pseudo-colored and shown in the figure. The color of pixels in these frames represents the value in the grayscale matrix extracted from the original frames. Pixels with grayscale exceeding 40 are displayed in white. The coordinate marked in these frames represents the row and column numbers of the grayscale matrix. The 3-m long gap occupies 170 rows in the grayscale rows, indicating the spatial resolution of these frames is about 0.0176 m per pixel. During the experiment, the relative humidity is 73 ± 2%, the temperature is 300 ± 0.21 K, the air pressure is 1 atm, and the calculated absolute humidity is 18.78 ± 0.51 g/m³.

average backward development velocity. The average bi-directional development velocity of the separate luminous structure is estimated to be less than $9 \times 10^5$ m/s and greater than $4.5 \times 10^5$ m/s, which is close to the leader-step formation speed of negative lightning leaders reported in recent literature[32].

**Comparison of leader steps of different polarities**. Eight consecutive frames recording a typical negative leader step are shown in Fig. 5. The negative leader discharge was produced in a rod-to-plane air gap with a 3 m length on August 31, 2021. Following the

negative leader step whose intense illumination was photographed in frame #1, a separate luminous structure was first captured in frame #3. The separate luminous segment emerged ahead of the leader channel tip and at about 0.53 m from the descending negative leader tip. Thereafter, the separate luminous structure's brightness progressively decreases until it is scarcely visible in frame #5. The separate luminous structure was captured again in frame #6. In frame #7, both the main leader channel and the separate luminous structure are significantly brighter, with the separate luminous structure appearing slightly brighter than the main negative leader channel. A streamer-like common zone

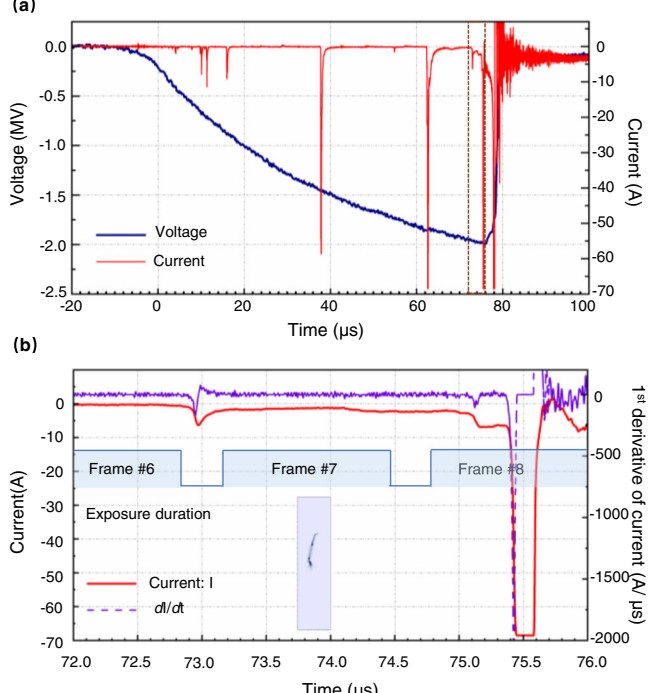

**Fig. 6 Discharge current associated with the negative leader step.** Panel **a** is the discharge current of the negative leader step, synchronized with the applied impulse voltage waveform. Panel **b** shows the detail of the current in the brown box marked in the above panel. The current is synchronized with the exposure strobe of the high-speed camera. The first derivative of the current was calculated, shown in the figure to indicate the variation of the current rise rate.

connecting the primary leader channel with the separate luminous structure, which was reported by previous studies[33], is obvious in the frame. The streamer-like common zone is around 0.38 m in length. The intense illumination of the merged leader channel captured by frame #8 indicates the occurrence of the negative leader step.

The current pulse of the negative leader step is shown in Fig. 6. Overall, the current pulse exhibits a sharp increase. The current pulse peak is less than −68 A, exceeding the measurement range. The current rose from −6.8 to −68 A in 73 nanoseconds. In detail, there is a stage during which the current is almost constant before abruptly rising. The period lasted about 2.55 μs. According to the synchronization of the current pulse and the high-speed camera frames shown in Fig. 4, this period includes the exposure duration of frame #7, which depicts the streamer-like common zone.

By comparing the high-speed camera frames (shown in Fig. 1) and current pulse waveform (shown in Fig. 2) associated with a positive leader step led by a separate luminous structure to the high-speed camera frames (shown in Fig. 5) and current pulse waveform (shown in Fig. 6) associated with a negative leader step, it is suggested that the positive leader step led by the luminous structure is accompanied by a steeply rising current pulse and strong illumination of the merged leader channel, similar to negative leader steps. This result is also consistent with previously described characteristics of positive leader steps[30].

## Discussion

According to the results illustrated in Figs. 1 and 2, the intense illumination of the merged leader channel and the sudden rise of the current pulse occurred following the abrupt elongation of the leader channel caused by the merging of a separate luminous

structure and the main leader channel. The mechanism of the strong illumination and rapid increase in current is of interest. Although no intense corona streamer burst was presented in the above results, in our view, a possible reason is that the intense corona streamer burst emerging from the merged leader channel tip may cause the steep current rise and intense illumination. In the literature reporting the separate luminous structure in the positive leader discharge, the intense corona streamer bursts in positive leader steps were described[30]. The fact that the intense corona streamer bursts were not observed may be due to the reason that the resolution and sensitivity of the high-speed camera without the image intensifier used in our experiments may not be sufficient to record the sudden development of streamers. Of course, at present, we cannot exclude other possibilities, the intense illumination and the sudden current rise also may be caused by the rapid-continuous development of the merged leader channel. More in-depth research is needed.

According to the previous literature[24], absolute humidity plays an important role in the occurrence of steps/restrikes in positive leader discharges. The results reported above were obtained at high absolute humidity (>18 g/m³). The effect of humidity on the emergence of the separate luminous structure leading positive leader steps needs further study.

It seems that the separate luminous structure in positive leader discharge and the space leader in the negative leader discharge plays a similar role in the formation of the steps of different polarities. This raises a critical question as to why the propagating positive leader tip is usually VHF silent. Recent literature indicated that it is actually the streamers in the intensive corona streamer burst that produce VHF[17,34]. The streamers in the intensive corona streamer burst occurring in positive leader steps are of the opposite polarity to the streamers in the intensive corona streamer burst occurring in negative leader steps. The polarity asymmetry of streamers may be the key to the question, which needs further research.

The above results imply that, at least at high absolute humidity, positive leaders can form steps due to the merging of a separate luminous structure and the primary leader channel, similar to the steps of negative leaders, which are formed due to merging the space leader with the main negative leader. These separate luminous structures appear in the leader corona zone ahead of the quasi-continuous propagating positive leader tip. The bi-directional development of the separate luminous structure and the streamer-like common zone that connects the main leader channel with the separate luminous structure was also observed. Both the current pulse of the positive leader steps led by the separate luminous structure and the current pulse of the negative leader steps led by the space leader is characterized by a sharp front. However, the physical nature and properties of the separate luminous structure leading positive leader steps need to be further researched.

## Methods

**Experimental conditions**. All records presented in this paper were obtained at the Ultra-High-Voltage Laboratory in Hefei, China. The experimental setups are shown in Fig. 7. The long spark discharges were produced in rod-to-plane air gaps by applying standard switching voltage impulses (with 250 μs front time and 2500 μs tail time), which were generated by a 4.8 MV Marx generator. The experiments were conducted in July 2020, August 2020, August 2021, and September 2021. In July 2020 and August 2020, the length of the air gaps was set as 4 m for positive sparks. The peak of the applied voltage impulses was set as 1.5 MV. In August 2021, the length of the air gaps was set as 3 m for negative sparks and the peak of the applied voltage impulses was set as 2.4 MV. In September 2021, the length of the air gaps was set as 5 m for positive sparks and the peak of the applied voltage impulses was set as 1.7 MV.

The control of the humidity is the key to the experiment. To observe the positive leader discharges at high humidity, the experiments were conducted in the summer of Hefei, in which the natural humidity is close to or even above 20 g/m³.

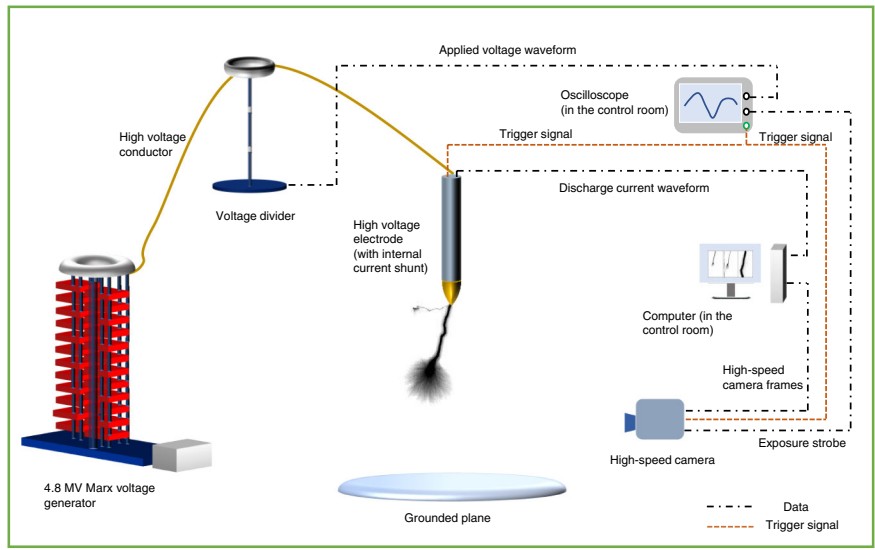

**Fig. 7 The experimental setups.** The experiments were conducted indoors. The length of the high voltage electrode is 2 m, and the gap scale is adjustable between 0 and 8 m. A voltage divider was used to measure the applied voltage waveform.

**Measurement.** A low-inductance shunt with an equivalent resistance of 5 Ω was mounted on the rod electrode to measure the discharge current. The main performance parameters of the current measurement system were as follows: −3 dB frequency bandwidth, DC to 18.3 MHz; sample rate, 2 GHz. High-speed cameras were used to record the development of leaders. The distance between the high-speed camera and the gap is varied for different gap lengths to guarantee that the field of view fulfills the requirement. To synchronize all experimental data in one discharge event, when the oscilloscope measuring the applied voltage waveforms was triggered, a 5 V TTL (Transistor-Transistor Logic) impulse generated by the oscilloscope was sent to the current measurement system and the high-speed video camera as a triggering signal. The trigger delay was pre-measured before the experiment.

The temperature and relative humidity during the experiments were measured by a HOBO MX1101 temperature/relative humidity data logger. The accuracy of the instrument is as follows: ±0.2 °C and ±2% RH. The absolute humidity was calculated from the relative humidity and temperature measured.

## Data availability

The datasets generated during and analyzed during the current study are available in the Zenodo database: https://zenodo.org/record/6597061.

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

## Acknowledgements

The authors are grateful for the funding support from State Grid Corporation of China (5442GY180059). The authors would like to thank three reviewers. The authors have benefited a lot from their comments and directions. In addition, the authors would like to thank Xia Chen for improving the language.

## Author contributions

S.H. drafted the manuscript. W.C. supervised the research. S.H., W.C., and N.X. revised the manuscript. S.H., Y.F., X.Q., N.X., and Z.Z. conducted the experiments and processed the data. Z.F., N.X., W.S., and D.C. contributed to the data analysis.

## Competing interests

The authors declare no competing interests.
