## [Peer Review File · Nature Communications]

Separate luminous structures leading positive leader stepsREVIEWER COMMENTS

Reviewer #1 (Remarks to the Author):

In “Polarity symmetry of leaders in moist air”, Huang et al. show new data from long spark measurements of positive leaders that show that positive leaders can produce space stems in high-humidity. Which is extremely important to the field and should be published. The authors even claim that the main current pulse during leader stepping is due to the corona flash and NOT due to the connection of the space stem and main channel as previously thought. It seems to me this observation is even MORE important than the current focus of the paper, and thus should be make a major point of the work IF the authors are able to present strong evidence of this viewpoint (which is presently not done, but perhaps could be).

However, despite the quality the data, the paper is very poorly written. The authors have a poor understanding of the field and regularly miss-quote references, sometimes even attributing the exact opposite conclusion that a reference actually gives. In particular, the authors seem to labor under the illusion that positive leader propagation emits copious VHF radiation and the source of this VHF is a mystery. This is precisely opposite the state of the field. It is well known that positive leader propagation is practically invisible in VHF, and it is other processes (needles and recoil leaders) that emit VHF from positive leader bodies. It is actually confusing why positive leader propagation emits so LITTLE VHF, and this work adds to that difficulty (which could also be a major point of the work). See ref 24 for a thorough review and discussion on how positive leader tips are VHF silent and that VHF emission from positive leaders is due to other processes.

The authors must correctly represent the field and the sources they are citing before this work can be published.

Many statements in the abstract and introduction are FAR to general:

Line 27: “The most important physics underlying lightning is the leader discharge”. To you or I perhaps, but this depends on what you care about. A structural engineer concerned about lightning protection, for example, only cares about return strokes.

Line 27-28: “The presence or absence of space stems/leaders in leader steps is the key to the polarity asymmetry of leaders,”. This again depends on perspective, particularly the scale of interest. I’d argue the difference between positive and negative streamers is key, as that is the underlying cause of space stems.

Line 34: “The lifetime of space stem/leader in positive leader steps is shorter than that in negative leader steps”. The authors should give numbers here.

Line 35: it is not clear what the authors mean by “causing the uncover of space stem/leader in previous studies”

Line 36-37: “The bidirectional development of space leaders in positive leader steps may be an important source for VHF radiations”. It is well known that positive leader tips produce very little to no VHF radiation. I discuss further below.

Line 37: “ outstanding problem that how positive lightning leaders produce VHF radiation.” Similarly, this is not an outstanding problem in the field. The real question is why positive leader propagation emits so little VHF. Again, I discuss further below.

Line 46: “had never discovered” → have never been observed

The statements from line 50 to line 54 are not correct. This paper claims that “ Copious radio electromagnetic pulses in the 30-300-megahertz frequency band are emitted during the negative leader steps attributed to the space leader's bidirectional development”, and cites references 4, 9, 15, 16, and 17. However, citation 4 claims that the RF radiations comes from the large current pulse that completes the stepping process (not space leader development). Reference 9 does not discuss RF emission at all. Reference 15 discusses HF (< 20 MHz), not VHF. Note that the difference between HF and VHF is critical and must be tracked carefully. This is because VHF frequencies are sensitive to meter-length scales and smaller where streamers are very important and HF frequencies are sensitive to 10 m and larger where long current pulses along the lightning channel are most important. Different frequencies see different things. Reference 16 is similar.

The typical view is that once the space leader connects to the main channel it produces a large current pulse (see ref 4, 9, 15) in order to equalize the potential, and this current pulse then produces a corona flash. It has previously been thought that the current pulse is the source of VHF [ref 4], but more recently it is thought that it is actually the streamers in the corona burst that produce VHF (see ref 17, and Hare et al. 2020 below).

On line 51-52 the authors claim that “here are varying reports regarding VHF emissions strength from positive leaders. In some literature, positive leaders are quiet at VHF/UHF[18], while some reports indicate that positive leaders radiate VHF rather more strongly than negative stepped leaders [19–23]”. This is completely incorrect and misleading. None of the references 19-23 say that VHF emission from

positive leader propagation is stronger than negative leaders. It is WELL accepted in the field that positive leader tips do not emit hardly any VHF. Refs 19 and 20 even point out that positive leader propagation (which they call streamers, as the nomenclature has changed over time), are invisible in VHF. I am disappointed that the authors don't reference 4 or 24 here, both of which discuss the fact that positive leaders emit very little to no VHF.

On line 53-54 the authors claim "It is puzzling that how positive leaders produce VHF radiation with the absence of space leaders[24]". This is LITERALLY the exact opposite point of ref 24, which discusses how the tip of the positive leader is nearly impossible to image in VHF due to how quiet it is.

The authors need to revise this work, here and elsewhere in the paper, so that it accurately represents the state of the field and that their claims actually represent their cites. The fact that the authors so misrepresent the literature and state of the field in this paragraph makes it very difficult for me to trust the rest of this work.

Lines 64-66 the authors claim "The brightness of the floating luminous formation is close to that of the primary leader channel, indicating that the temperature inner the floating luminous formation may also be close to the leader core's temperature.". This is incorrect, as it assumes that this formation is thermalized. If the plasma hasn't reached thermal equilibrium then the concept of temperature does not even apply. Malagón-Romero and Luque (2019) even discuss how such luminous spots could occur without being at thermal equilibrium.

Both figures 2 and 3 show current vs time and a series of photos. However, it is very difficult to interpret this data as there is no indication of the time periods that were integrated to produce the photos. This could possibly be done by showing vertical bars in the current vs time (somewhat like the middle bit of fig. 1) .

The section starting on line 86 discusses the authors interpretation of their data that the large current pulse is due to the streamer burst and not due to the space leader connecting with the channel. I actually like this interpretation, as it is very similar to a conclusion my group has come to recently with different data. However, it is VERY different from the standard model accepted by the community (I discuss above), that the large current pulse is due to the space leader connecting with the main channel. This difference from the commonly accepted model needs to be stated and discussed clearly, and strong evidence needs to be presented to support why the authors believe this (as presently very little evidence is given). One possibility is to modify figures 2 and 3 to show how the time of the high-speed frames with reference to current, and to demonstrate that the current pulse occurs significantly after the space leader/main channel connection, during the corona burst. This, however, could be very difficult to actually show.

If the authors ARE able to put strong evidence behind the possibility that the main current pulse is due to the corona burst and NOT the space leader/main channel connection, then this should be made a very important key point of the paper as it is extremely important to the field and very different from the common interpretation.

Line 112 leade → leader

Line 143 refers to “needle-like discharge”. It is not clear what the authors are referring to here. If the authors are referring to the discovery in ref 24, then ref 24 needs to be cited here.

Line 146 needs to cite ref 17.

Lines 147 and below discuss humidity. However, the authors do not discuss how the humidity was measured or controlled (was it natural variation or was there artificial humidity control?) in the methods section. Line 149 is inconsistent with line 147, in that line 149 discusses experiments with much lower humidity. Were these experiments part of a different publication? In which case a citation needs to be given. If not, then a thorough description of the measurements needs to be given. Finally, the authors state that lower humidity results in less space stem formation. Numbers need to be given here, perhaps a ratio of sparks with and without space stem formation for both lower and higher humidities.

The following claims need citations:

line 156: strong electronegativity can accelerate the electron attachment process

Line 157: as humidity increases the internal electric field in a positive streamer is enhanced

line 159: when electrons flow along the stem only a fraction of energy loss is transferred to thermal energy

line 161: a larger fraction goes to vibrational excitation

line 163 the relaxation time is dependent on humidity

It is likely I have missed other claims that need citations as well. This should be fixed throughout the paper.

The method section is too brief. Besides a discussion on humidity measurement and control, the authors also need to explain how they generated their long spark. What the peak voltage was, the length of the sparks, and the voltage ramp up and ramp down times.

Hare et al. 2020, "Radio Emission Reveals Inner Meter-Scale Structure of Negative Lightning Leader Steps", doi: 10.1103/PhysRevLett.124.105101

A. Malagón-Romero and A. Luque (2019), "Spontaneous emergence of space stems ahead of negative leaders in lightning and long sparks". *Geophys. Res. Lett.* 46, 4029

Reviewer #2 (Remarks to the Author):

The manuscript by Chen et al. presents what are, to my knowledge, the first images of a distinct space stem ahead of an advancing positive leader. This is a remarkable result that potentially deserves to be published in *Nature Communications*.

However in my view the paper, as it stands now, has several shortcomings that prevent me from recommending its publication. I suggest that the authors work to improve substantially the presentation of their results before they re-submit this manuscript.

We may start with the manuscript's title, "Polarity symmetry of leaders in moist air". This is quite confusing as the authors know well that the dynamics of positive and negative leaders is quite different even in moist air. This can be concluded from their own data: whereas they recorded many bi-directional steps negative leaders, they found only one example in a positive leader.

In general the paper also is lacking in details and the "Methods" section is clearly insufficient. Ideally the authors should provide enough details to let other laboratories replicate their results. But this is far from the case here: we do not get to know the wiring of their setup or the details of the power switch. What is the rise-time, the peak applied voltage, how stable is their configuration, what was the repetition rate of the discharges? All these details are relevant to put the presented observation in the context of current research. Instead in the Methods section we find an unuseful comparison to the Apollo project and a short historical digression about the value of laboratory observations of long sparks.

The authors should also revise their hypothesis that in summer thunderstorms the emission of VHF radiation from positive or negative leaders should be similar. This is contradicted by many years of Lightning Mapping Array (LMA) observations, which show clear differences between positive and negative leaders.

Reviewer #3 (Remarks to the Author):

Please see attached file.

Review 1

<1> The most important result of the article is the expansion in two directions of the quasi-space leader in the streamer crown of a positive long spark and the possible formation of a step. Strictly speaking, the authors did not receive an image of the step, since the main stage of the long spark breakdown is already visible in frame No. 9. The authors first discovered bidirectional propagation of a quasi-space-leader of a positive spark, but a quasi-space leader in a long spark was first discovered in the article (Kostinskiy et al., 2018, Figure 10b). The reviewer does not recommend writing a space leader, since the space leader is a hot plasma formation discovered long ago in the streamer corona of a negative long spark. From the available experiments, we cannot yet establish the same nature of the plasma of the quasi-space leader, discovered in the article (Kostinskiy et al., 2018) and in this study, and the space leader of the negative spark.

<2> Authors need to give a curve of camera matrix sensitivity depending on the wavelength of light. Your camera should almost not be able to sense UV radiation, as can be clearly seen in Figures 1-3. I gave an example of the usual curve of the camera matrix sensitivity below. Most speed cameras have similar curves that are not sensitive to UV light.

<3> **Lines 32-34.** *Here we report the emergence of the space stem and the bidirectional development of the space leader in positive leader steps in moist air, using a high-speed camera with unprecedented spatial-temporal resolution.*

This statement is not correct, since many articles achieve much better spatial and temporal resolution less than 50 ns and with a size less than 1 mm (for example, Kochkin et al., 2012; Kostinskiy et al., 2018)

<4> It is desirable that the authors clearly distinguish in their work what they themselves saw in their experiments, and what they assume, relying on physical models that are in scientific articles and books. At the same time, it is desirable that each thought from other articles has a reference.

<5> The entire text of the paper contains a serious incorrectness, when the authors use the term: space stems/leaders Based on the physics of a long spark, we can conclude that these are completely different plasma formations (Stekolnikov and Shkilyov, 1963; Les Renardieres Group, 1981, Gorin et al., 1976; Bazelyann and Raizer, 1998; Rakov and Uman, 2003). And the results of the study of lightning do not yet give a clear answer. In the streamer zone of a long negative spark, researchers have long identified two different plasma formations: a space-stem and a space-leader, and they never mix the two. When studying lightning, scientists cannot yet firmly say what they see in the streamer zone of a negative and positive leader, and therefore they write space stems/leaders. Scientists write this not because they believe that it is one and the same plasma formation, but because they do not know exactly what they are seeing.

Lines 45-46: *Positive leaders usually propagate continuously but also exhibit steps under particular conditions*⁸⁻¹⁴.

This is an erroneous statement. Positive leaders in a quasi-continuous phase always move intermittently, but their steps are the size of a positive leader's head (Bazelyan and Raizer, 1998; Popov, 2009).

Lines 46-47: *Space stems/leaders had never discovered in positive leader steps.*

This is an incorrect statement. The article (Kostinskiy et al., 2018, Figure 10b) directly shows a quasi-space leader with a stepwise propagation of a positive leader (restrikes). This discovery is unambiguously written several times in the article and there is a special section where Figure 10b is discussed in detail.

Lines 54-55, 61-62, 68-69, 131-132: *Here we report the emergence of space stems and the bi-directional development of space leaders in positive leader steps.*

The authors most likely in this place of the text and in the rest of the text cannot clearly identify space stems on their frames, since the matrix of a high-speed camera will not allow this to be done.

Lines 56-58: *Our results suggest that high humidity during summer thunderstorms will break the polarity asymmetry of lightning leaders and illuminates insight into how positive lightning leaders produce VHF radiation.*

This phrase needs to be changed, since the results of the authors confirmed the long-standing result of the Les Renardieres Group (Les Renardieres Group, 1977).

Lines 70-71: *The estimated backward velocity of the space leader is larger than 1.58×10^5 m/s. The estimated forward velocity of the space leader is about 3 times the value, significantly larger than the velocity of continuously developing positive leaders.*

1.58×10^5 m/s is an unrealistically high speed for the development of a quasi-space leader, usually the propagation speed is much lower -- 1-3 cm/ μ s ($1-3 \times 10^4$ m/s). Therefore, it is desirable that the authors indicate the procedure for measuring the speed: what points on which frames did the authors take to measure the speed. The average quasi-continuous

propagation velocity of the leader (at front 100-250 μs) is about 1-2 cm/ μs , and this is about 10 times less than the speed of 1.58×10^5 m/s.

Lines 75-77: *Thus, we infer that the space stem/leader lifetime (defined as the time from the emergence of the space stem to the full integration of the space leader into the primary leader channel in this letter) is less than 2.22 μs in the event.*

This phrase needs to be corrected, as the authors must prove that they see the quasi-space-stem.

Figure 1. What do the numbers (10-120, 10-50) mean in color illustrations? It is necessary to set the dimensions in the Figure so that the readers can themselves estimate the velocities and sizes of the plasma formations. For Figure 1, the authors also need to give a current oscillogram, as for Figure 2, as it will greatly complement the information that relates to this unique phenomenon.

Lines 83-84: *The lightning-like discharge is produced by applying the standard positive switching impulse (SI) to a 4 m point-to-plane gap in the summer of 2020.*

It is desirable that the authors give the exact value of the front and the duration of the voltage pulse (250/3500 μs ?), Since not all readers of the article work in the field of high-voltage technology.

Lines 84-85: *The absolute humidity is about 20 g/m³, the air pressure is 1 atm.*

With what device and with what error did the authors measure humidity? Humidity is a very important parameter for the appearance of positive leader steps.

Figure 2. It is desirable that the authors synchronize the frames and the current and give the entire oscillogram of the current and voltage that correspond to this phenomenon. What do the numbers (10-120, 10-50) mean in color drawings? It is necessary to supply the dimensions in the figure. The authors again incorrectly use the term space-stem, since they have no experimental evidence that this plasma formation is precisely a space-stem. Without a clear image of the streamer zone, it is impossible to separate the space stem from the space leader. Most likely a space leader is depicted here. In case of difficulties with the identification of plasma objects, it is reasonable for the authors to write "plasma object" or "plasma formation". The 169 ns figure is printed in the wrong place, as it falls on the dotted line.

Lines 93-95: *We infer that the current rise in this stage corresponds to the collision of opposite-polarity streamers emerging from the descending leader channel tip and the space stem, respectively.*

What are the authors' grounds for such assumptions and inventing a new term (collision)? The breakthrough phase of interaction between the leaders, with the exception of the first few nanoseconds, is carried out by positive streamers, since the electric field required for the movement of positive streamers is two times less than for negative ones. If the authors do not have evidence of some facts, then it is advisable not to write them. Authors can write their guesses in the discussion of experimental results.

Lines 96-98: *When the space leader merges with the primary positive leader, the primary positive leader tip's high potential is transferred to the lower end of the space leader, followed by an intense positive leader corona streamers burst reported by previous literature.*

Dear colleagues, if you refer to literature, it is always necessary to indicate specific articles and books.

Lines 100: *The third stage is referred to as the decay stage here, lasting several microseconds.*
The text does not make it clear to the reader what exactly is disintegrating? If in the experiment the voltage pulse lasted for several ms, then the plasma decay will also last for several ms, and the authors should see the plasma decay in many hundreds of subsequent frames.

Lines 101-103: *Figure 3 shows the current pulse of a negative leader step led by a space stem and 6 consecutive frames recording the step. The space stem emerged at 0.4 m from the descending leader channel tip. The space stem/lifetime is longer than 4.44 μ s.*

The authors incorrectly write the term space-stem, since the analysis of Figure 3 does not allow us to establish this. The authors can only correctly assume that this is a space-stem. Also, the authors cannot accurately say the lifetime of the space stem if they cannot accurately establish the nature of the plasma object.

Lines 103-104: *The space leader's estimated backward velocity is larger than 6.7×10^5 m/s, which is close to the estimated forward velocity of the space leader shown in Figure 1.*

It is desirable that the authors specify exactly what backward velocity is and describe in detail the procedure for measuring the velocity: which points on which frames and what time interval the authors used to measure the velocity. And everywhere it is desirable that the authors indicate that they measure the average speed over a certain period of time.

Lines 106-107: *The difference is that the collision stage duration of the negative leader step (lasting about 3.5 μ s) is obviously longer than that of the positive leader step led by the space stem*

The long spark literature does not include such a term as the collision stage. The literature contains such a well-known term as - end-to-end phase. If the authors introduce a new term, then they must have very good reasons, which they must detail in the text.

Lines 113-114: *That may indicate that the space stem/leader lifetime in positive leader steps tends to be shorter than in negative leader steps.*

This statement is not correct and must be removed, since this conclusion does not follow from experiments.

Lines 131-132. *The results reported here powerfully proves the existence of space stem/leader in positive leader steps, ending a long-standing argument.*

It is desirable that the authors change this phrase. The paper produced an important result, shown in Figure 1, but the dispute not only does not end, but only begins. One single experiment cannot serve as definitive proof.

Lines 134-135. *The intense leader corona streamers burst occurring after the space leader merges with the primary positive leader injects a strong current pulse into the leader channel.*

This phrase needs to be changed. Figures 2,3 do not allow to distinguish streamer flashes from the powerful illumination of the camera matrix, which occurs during the step. Figure 1 clearly shows that the speed camera does not reliably identify streamers.

Lines 143-144. *VHF radiation from the positive lightning leaders is still not well understood. The needle-like discharge that may occur after positive leader development pausing is identified as a critical source.*

It is advisable that the authors make a reference to the paper they are discussing. Need to explain what a critical source is?

Lines 150-151. *That indicates the absolute humidity may play an important role in the emergence of the space stem/leader in positive leader steps.*

This conclusion was finally made in 1977 by (Les Renardieres Group (1977)).

Lines 166-180. Method. This work is experimental. Therefore, it is desirable that the authors describe in detail the measurement methods, and not give reasoning on general topics. It is desirable that the authors give: the parameters of the bandwidth of the oscilloscope, complete oscillograms of the current (at the high-voltage electrode?) And the discharge voltage, the parameters of the measuring shunt, the specific brand of the high-speed camera, the sensitivity of the camera matrix depending on the wavelength. The general scheme of the experiment and at what distance the camera was from the discharge would be very useful to the readers. How and with what error were measured humidity, temperature, etc.

Lines 171-172. *The space stem/leader leading to negative leader steps was first discovered in the laboratory³⁵⁻³⁷ and later confirmed through natural lightning observations³⁸.*

This phrase needs reworking. For the first time, the structure of the streamer crown of the negative leader of a long spark was established Stekolnikov and Shkilev, (1963). The article by Mitchell and Snoddy (1947) has nothing to do with a long spark and should be deleted as the experiments were carried out in tubes (not free space) at very low pressure and low voltage.

“Properties of the progressive breakdown, occurring when an impulsive potential is applied to an electrode in one end of a discharge **tube 14 cm in diameter and 12 meters long**, have been investigated, principally by means of a high speed cathode-ray oscillograph. Potentials from **25 kv to 115 kv and pressures from 0.006 to 8.0 mm Hg** were used, with dry air and hydrogen in the tube” (Mitchell and Snoddy, 1947, <https://doi.org/10.1103/PhysRev.72.1202>).

References

Bazelyan, E. M., & Raizer, Y. P. (1998). Spark discharge (p. 294). Boca Raton, FL: CRC Press.

Gorin, B. N., Levitov, V. I., & Shkilev, A. V. (1976). Some principles of leader discharge of air gaps with a strong non-uniform field. IEE Conference Publication, 143, 274–278.

Kochkin, P O, C V Nguyen, A P J van Deursen and U Ebert (2012), Experimental study of hard X-rays emitted from meter-scale positive discharges in air, arXiv:1208.5899v2 [physics.plasm-ph] 13 Sep 2012

Kostinskiy, A. Y., Syssoev, V. S., Bogatov, N. A., Mareev, E. A., Andreev, M. G., Bulatov, M. U., et al. (2018). Abrupt elongation (stepping) of negative and positive leaders culminating in an intense corona streamer burst: Observations in long sparks and implications for lightning. *Journal of Geophysical Research: Atmospheres*, 123. <https://doi.org/10.1029/2017JD027997>

Les Renardieres Group (1977). Positive discharges in long air gaps at Les Renardieres, 1975 results and conclusions. *Electra*, 53, 31–153.

Rakov, V. A., & Uman, M. A. (2003). *Lightning: Physics and effects* (p. 687). New York: Cambridge University Press. <https://doi.org/10.1017/CBO9781107340886>

Popov, N. A. (2009). Study of the formation and propagation of a leader channel in air. *Plasma Physics Reports*, 35(9), 785–793. <https://doi.org/10.1134/S1063780X09090074>

Stekolnikov I.S. and Shkilyov A.V. (1963). The development a long spark and lightning. *Proc. Of the Third I.C. of Atmosphere and Space electricity*. Montreux, Switzerland, may, 5-10, pp.466-481

Response to Reviewers

We would first like to express our gratitude to the three reviewers. We have carefully reviewed the comments. The comments from two reviewers are very important for us to improve the manuscript. Under the guidance of the reviewers, we realized that the previous version has serious problems and is not a rigorous scientific paper. We are sorry! We have revised the manuscript accordingly. Our responses are given in a point-by-point manner below.

Response to Reviewer #1

Comment: In “Polarity symmetry of leaders in moist air”, Huang et al. show new data from long spark measurements of positive leaders that show that positive leaders can produce space stems in high-humidity. Which is extremely important to the field and should be published. The authors even claim that the main current pulse during leader stepping is due to the corona flash and NOT due to the connection of the space stem and main channel as previously thought. It seems to me this observation is even MORE important than the current focus of the paper, and thus should be make a major point of the work IF the authors are able to present strong evidence of this viewpoint (which is presently not done, but perhaps could be).

Response: Under the direction of the comment, we realize that the physics underlying the current pulse of positive leader steps led by the separate luminous structure is well worth an in-depth study. As suggested by the reviewer, we conducted complementary experiments in September 2021. In one event, a high-speed camera frame showing that the separate luminous structure connects the primary leader channel through a like-common-streamer-zone was obtained. During the exposure duration of the frame, the channel current rose slowly. The data may not be strong evidence, but we thought that the data may deepen the understanding of the physics underlying the current pulse of steps. The data is shown in Figure 1, Figure 2, and Figure 3 in the revised manuscript.

Comment: However, despite the quality the data, the paper is very poorly written. The authors have a poor understanding of the field and regularly miss-quote references, sometimes even attributing the exact opposite conclusion that a reference actually gives. In particular, the authors seem to labor under the illusion that positive leader propagation emits copious VHF radiation and the source of this VHF is a mystery. This is precisely opposite the state of the field. It is well known that positive leader propagation is practically invisible in VHF, and it is other processes (needles and recoil leaders) that emit VHF from positive leader bodies. It is actually confusing why positive leader propagation emits so LITTLE VHF, and this work adds to that difficulty (which could also be a major point of the work). See ref 24 for a thorough review and discussion on how positive leader tips are VHF silent and that VHF emission from positive leaders is due to other processes. The authors must correctly represent the field and the sources they are citing before this work can be published.

Response: Thanks to the reviewer for pointing out the serious problem. Under the guidance of the reviewer and reviewer #2, we realized that our previous understanding of the VHF radiation from positive leaders was significantly mistaken. We have reread the paper cited and tried our best to eliminate and avoid incorrect citations. Thanks to the reviewer for helping us to build up the right understanding of the issue.

Comment: Many statements in the abstract and introduction are FAR too general:
Line 27: "The most important physics underlying lightning is the leader discharge". To you or I perhaps, but this depends on what you care about. A structural engineer concerned about lightning protection, for example, only cares about return strokes.
Line 27-28: "The presence or absence of space stems/leaders in leader steps is the key to the polarity asymmetry of leaders,". This again depends on perspective, particularly the scale of interest. I'd argue the difference between positive and negative streamers is key, as that is the underlying cause of space stems.

Response: We have realized the problem. The inappropriate statement has been corrected in the revised manuscript. Thanks!

Comment: Line 34: "The lifetime of space stem/leader in positive leader steps is shorter than that in negative leader steps". The authors should give numbers here.

Response: Under the guidance of reviewer #3, we realized that the definition of *the lifetime of space stem/leader* lacked a scientific basis. We have deleted the sentence in the revised manuscript. Thank you and reviewer #3!

Comment: Line 35: it is not clear what the authors mean by "causing the uncover of space stem/leader in previous studies"

Response: The inappropriate statement has been deleted in the revised manuscript. Thanks!

Comment: Line 36-37: "The bidirectional development of space leaders in positive leader steps may be an important source for VHF radiations". It is well known that positive leader tips produce very little to no VHF radiation. I discuss further below.

Response: We have realized the serious problem. The incorrect statement has been deleted in the revised manuscript. Thanks!

Comment: Line 37: "outstanding problem that how positive lightning leaders produce VHF radiation." Similarly, this is not an outstanding problem in the field. The real question is why positive leader propagation emits so little VHF. Again, I discuss further below.

Response: We have realized the serious problem. The serious problem has been solved in the revised manuscript. Thanks!

Comment: Line 46: “had never discovered” → have never been observed

Response: The problem has been solved in the revised manuscript. Thanks!

Comment: The statements from line 50 to line 54 are not correct. This paper claims that “ Copious radio electromagnetic pulses in the 30-300-megahertz frequency band are emitted during the negative leader steps attributed to the space leader's bidirectional development”, and cites references 4, 9, 15, 16, and 17. However, citation 4 claims that the RF radiations comes from the large current pulse that completes the stepping process (not space leader development). Reference 9 does not discuss RF emission at all. Reference 15 discusses HF (< 20 MHz), not VHF. Note that the difference between HF and VHF is critical and must be tracked carefully. This is because VHF frequencies are sensitive to meter-length scales and smaller where streamers are very important and HF frequencies are sensitive to 10 m and larger where long current pulses along the lightning channel are most important. Different frequencies see different things. Reference 16 is similar.

Response: We have realized the serious problem. The serious problem has been solved in the revised manuscript. Thank the reviewer very much!

Comment: The typical view is that once the space leader connects to the main channel it produces a large current pulse (see ref 4, 9, 15) in order to equalize the potential, and this current pulse then produces a corona flash. It has previously been thought that the current pulse is the source of VHF [ref 4], but more recently it is thought that it is actually the streamers in the corona burst that produce VHF (see ref 17, and Hare et al. 2020 below).

Response: Thank the guidance of the reviewer. The comment helped us to build up the right understanding. In our view, the space leader is first connected to the main leader channel through a common streamer zone with high impedance. The potential transfer between the main leader tip and the bottom end of the space leader may not immediately result in a large channel current pulse. In the exposure duration of frame #4 shown in Figure 1 in the revised manuscript, the separated luminous structure connected the main leader tip through a like-common-streamer-zone, but the channel current rose slowly.

Comment: On line 51-52 the authors claim that “here are varying reports regarding VHF emissions strength from positive leaders. In some literature, positive leaders are quiet at VHF/UHF[18], while some reports indicate that positive leaders radiate VHF rather more strongly than negative stepped leaders [19–23]”. This is completely incorrect and misleading. None of the references 19-23 say that VHF emission from positive leader propagation is stronger than negative leaders. It is WELL accepted in the field that positive leader tips do not emit hardly any VHF. Refs 19 and 20 even point out that positive leader propagation (which they call streamers, as the nomenclature has changed over time), are invisible in VHF. I am disappointed that the authors don't reference 4 or 24 here, both of which discuss the fact that positive leaders emit very little to no VHF.

Response: We have realized the serious problem. The serious problem has been solved in the revised manuscript. Thanks!

Comment: On line 53-54 the authors claim “It is puzzling that how positive leaders produce VHF radiation with the absence of space leaders[24]”. This is LITERALLY the exact opposite point of ref 24, which discusses how the tip of the positive leader is nearly impossible to image in VHF due to how quiet it is.

Response: We have corrected the serious problem in the revised manuscript. Thanks!

Comment: The authors need to revise this work, here and elsewhere in the paper, so that it accurately represents the state of the field and that their claims actually represent their cites. The fact that the authors so misrepresent the literature and state of the field in this paragraph makes it very difficult for me to trust the rest of this work.

Response: Thank the reviewer very much! We have tried our best to correct the serious problem in the revised manuscript.

Comment: Lines 64-66 the authors claim “The brightness of the floating luminous formation is close to that of the primary leader channel, indicating that the temperature inner the floating luminous formation may also be close to the leader core's temperature.”. This is incorrect, as it assumes that this formation is thermalized. If the plasma hasn't reached thermal equilibrium then the concept of temperature does not even apply. Malagón-Romero and Luque (2019) even discuss how such luminous spots could occur without being at thermal equilibrium.

Response: Under the direction of the comment, we have realized the problem and have deleted the incorrect statement in the revised manuscript.

Comment: Both figures 2 and 3 show current vs time and a series of photos. However, it is very difficult to interpret this data as there is no indication of the time periods that were integrated to produce the photos. This could possibly be done by showing vertical bars in the current vs time (somewhat like the middle bit of fig. 1) .

Response: We have revised the figure according to the direction of the comment in the revised manuscript. Thanks!

Comment: The section starting on line 86 discusses the authors interpretation of their data that the large current pulse is due to the streamer burst and not due to the space leader connecting with the channel. I actually like this interpretation, as it is very similar to a conclusion my group has come to recently with different data. However, it is VERY different from the standard model accepted by the community (I discuss above), that the large current pulse is due to the space leader connecting with the main channel. This difference from the commonly accepted model needs to be stated and discussed clearly, and strong

evidence needs to be presented to support why the authors believe this (as presently very little evidence is given). One possibility is to modify figures 2 and 3 to show how the time of the high-speed frames with reference to current, and to demonstrate that the current pulse occurs significantly after the space leader/main channel connection, during the corona burst. This, however, could be very difficult to actually show.

If the authors ARE able to put strong evidence behind the possibility that the main current pulse is due to the corona burst and NOT the space leader/main channel connection, then this should be made a very important key point of the paper as it is extremely important to the field and very different from the common interpretation.

Response: According to the comment, in figure 3 in the revised manuscript, the exposure strobe signal of the high-speed camera frames recording the step and the step current pulse. On this basis, we have tried to discuss the physics underlying the step current pulse in the revised manuscript from line 171 to line 188. The comment is constructive. Thanks !

Comment: Line 112 leade → leader

Response: Thanks ! We have corrected the problem.

Comment: Line 143 refers to “needle-like discharge”. It is not clear what the authors are referring to here. If the authors are referring to the discovery in ref 24, then ref 24 needs to be cited here.

Response: We have corrected the problem in the revised manuscript.

Comment: Line 146 needs to cite ref 17.

Response: We have corrected the problem in the revised manuscript. Thanks!

Comment: Lines 147 and below discuss humidity. However, the authors do not discuss how the humidity was measured or controlled (was it natural variation or was there artificial humidity control?) in the methods section. Line 149 is inconsistent with line 147, in that line 149 discusses experiments with much lower humidity. Were these experiments part of a different publication? In which case a citation needs to be given. If not, then a thorough description of the measurements needs to be given. Finally, the authors state that lower humidity results in less space stem formation. Numbers need to be given here, perhaps a ratio of sparks with and without space stem formation for both lower and higher humidity.

Response: The measurement and control of the humidity are the keys to the experiment. Artificial humidity control is difficult for meter scale spark discharges. We conducted the experiments at the natural high humidity environment. According to the previous literature, the steps are absent in positive leader discharges under dry atmospheric conditions. We have corrected the serious problem in the revised manuscript from line 245 to 247 in the revised manuscript. Thanks!

Comment: The following claims need citations:

line 156: strong electronegativity can accelerate the electron attachment process

Line 157: as humidity increases the internal electric field in a positive streamer is enhanced

line 159: when electrons flow along the stem only a fraction of energy loss is transferred to thermal energy

line 161: a larger fraction goes to vibrational excitation

line 163 the relaxation time is dependent on humidity

It is likely I have missed other claims that need citations as well. This should be fixed throughout the paper.

Response: Thank the reviewer for pointing the problem! We have corrected these problems in the revised manuscript.

Comment: The method section is too brief. Besides a discussion on humidity measurement and control, the authors also need to explain how they generated their long spark. What the peak voltage was, the length of the sparks, and the voltage ramp up and ramp down times.

Response: Under the guidance of three reviewer, we have rewritten the **Method** section in the revised manuscript.

Thank the reviewer very much!

Response to Reviewer #2

The manuscript by Chen et al. presents what are, to my knowledge, the first images of a distinct space stem ahead of an advancing positive leader. This is a remarkable result that potentially deserves to be published in Nature Communications.

However in my view the paper, as it stands now, has several shortcomings that prevent me for recommending its publication. I suggest that the authors work to improve substantially the presentation of their results before they re-submit this manuscript.

Comment: We may start with the manuscript's title, "Polarity symmetry of leaders in moist air". This is quite confusing as the authors know well that the dynamics of positive and negative leaders is quite different even in moist air. This can be concluded from their own data: whereas they recorded many bi-directional steps negative leaders, they found only one example in a positive leader.

Response: The comment is important for us. We realized the problem under the guidance of the reviewer. We have revised the title of the manuscript.

Comment: In general the paper also is lacking in details and the "Methods" section is clearly insufficient. Ideally the authors should provide enough details to let other

laboratories replicate their results. But this is far from the case here: we do not get to know the wiring of their setup or the details of the power switch. What is the rise-time, the peak applied voltage, how stable is their configuration, what was the repetition rate of the discharges? All these details are relevant to put the presented observation in the context of current research. Instead in the Methods section we find an unuseful comparison the Apollo project and a short historical digression about the value of laboratory observations of long sparks.

Response: That is a serious problem. we have rewritten the **Method** section in the revised manuscript under the guidance of three reviewers. Thank the reviewer very much!

Comment: The authors should also revise their hypothesis that in summer thunderstorms the emission of VHF radiation from positive or negative leaders should be similar. This is contradicted by many years of Lightning Mapping Array (LMA) observations, which show clear differences between positive and negative leaders.

Response: Under the guidance of the reviewer and reviewer #1, we have realized the serious problem and have corrected the problem in the revised manuscript.

Thank the reviewer very much!

Response to Reviewer #3

Comment: The most important result of the article is the expansion in two directions of the quasi-space leader in the streamer crown of a positive long spark and the possible formation of a step. Strictly speaking, the authors did not receive an image of the step, since the main stage of the long spark breakdown is already visible in frame No. 9. The authors first discovered bidirectional propagation of a quasi-space-leader of a positive spark, but a quasi-space leader in a long spark was first discovered in the article (Kostinskiy et al., 2018, Figure 10b). The reviewer does not recommend writing a space leader, since the space leader is a hot plasma formation discovered long ago in the streamer corona of a negative long spark. From the available experiments, we cannot yet establish the same nature of the plasma of the quasi-space leader, discovered in the article (Kostinskiy et al., 2018) and in this study, and the space leader of the negative spark.

Response: The separated luminous structure in positive leader discharges was first discovered by Kostinskiy et al, suggesting the positive leaders may do steps in a similar way to negative leaders under certain conditions. In our view, the date shown in Figure 2 in the revised manuscript could prove that the separated luminous structure can indeed lead positive leader steps, and the date shown in Figure 4 in the revised manuscript indicates the bi-directional development of the separated luminous structure. The comment is important for us and the problem pointed by the reviewer has been corrected in the revised manuscript. We write the *separated luminous structure* instead of *space leader* in the revised manuscript. Thank the reviewer very much!

Comment: Authors need to give a curve of camera matrix sensitivity depending on the wavelength of light. Your camera should almost not be able to sense UV radiation, as can be clearly seen in Figures 1-3. I gave an example of the usual curve of the camera matrix sensitivity below. Most speed cameras have similar curves that are not sensitive to UV light.

Response: Thank the reviewer for pointing the problem. The color in the pseudo-color images only represents the grayscale. This point was not made clear to readers in the previous manuscript. We have corrected the problem in the revised manuscript. In addition, the models of manufacturers of the high-speed cameras have been added in the **Method** section in the revised manuscript. Thanks!

Comment: Lines 32-34. Here we report the emergence of the space stem and the bidirectional development of the space leader in positive leader steps in moist air, using a high-speed camera with unprecedented spatial-temporal resolution.

This statement is not correct, since many articles achieve much better spatial and temporal resolution less than 50 ns and with a size less than 1 mm (for example, Kochkin et al., 2012; Kostinskiy et al., 2018)

Response: We realized the problem! The problem has been corrected in the revised manuscript. Thanks!

Comment: It is desirable that the authors clearly distinguish in their work what they themselves saw in their experiments, and what they assume, relying on physical models that are in scientific articles and books. At the same time, it is desirable that each thought from other articles has a reference.

Response: We are sorry for that there are many problems pointed by three reviewers. Under the guidance of the reviewers, we have carefully revised the manuscript.

Comment: The entire text of the paper contains a serious incorrectness, when the authors use the term: space stems/leaders Based on the physics of a long spark, we can conclude that these are completely different plasma formations (Stekolnikov and Shkilyov, 1963; Les Renardieres Group, 1981, Gorin et al., 1976; Bazelyann and Raizer, 1998; Rakov and Uman, 2003). And the results of the study of lightning do not yet give a clear answer. In the streamer zone of a long negative spark, researchers have long identified two different plasma formations: a space-stem and a space-leader, and they never mix the two. When studying lightning, scientists cannot yet firmly say what they see in the streamer zone of a negative and positive leader, and therefore they write space stems/leaders. Scientists write this not because they believe that it is one and the same plasma formation, but because they do not know exactly what they are seeing.

Response: We have realized the serious problem under the direction of the comment. We have replaced the *space leader/space stem* with *the separated luminous structure* in the revised manuscript. Thank the reviewer very much!

Comment: Lines 45-46: Positive leaders usually propagate continuously but also exhibit steps under particular conditions⁸⁻¹⁴ .

This is an erroneous statement. Positive leaders in a quasi-continuous phase always move intermittently, but their steps are the size of a positive leader's head (Bazelyan and Raizer, 1998; Popov, 2009)

Response: We have corrected the problem in the revised manuscript! Thank the reviewer for pointing the serious problem!

Comment: *Space stems/leaders had never discovered in positive leader steps.*

This is an incorrect statement. The article (Kostinskiy et al., 2018, Figure 10b) directly shows a quasi-space leader with a stepwise propagation of a positive leader (restrikes). This discovery is unambiguously written several times in the article and there is a special section where Figure 10b is discussed in detail.

Response: We have corrected the problem in the revised manuscript! Thank the reviewer very much!

Comment: *Lines 54-55, 61-62, 68-69, 131-132: Here we report the emergence of space stems and the bidirectional development of space leaders in positive leader steps.*

The authors most likely in this place of the text and in the rest of the text cannot clearly identify space stems on their frames, since the matrix of a high-speed camera will not allow this to be done.

Response: The problem has been corrected in the revised manuscript. Thanks!

Comment: *Our results suggest that high humidity during summer thunderstorms will break the polarity asymmetry of lightning leaders and illuminates insight into how positive lightning leaders produce VHF radiation.*

This phrase needs to be changed, since the results of the authors confirmed the long-standing result of the Les Renardieres Group (Les Renardieres Group, 1977)

Response: We have corrected the problem in the revised manuscript under the direction of the comment. Thanks!

Comment: *The estimated backward velocity of the space leader is larger than 1.58×10^5 m/s. The estimated forward velocity of the space leader is about 3 times the value, significantly larger than the velocity of continuously developing positive leaders.*

1.58×10^5 m/s is an unrealistically high speed for the development of a quasi-space leader, usually the propagation speed is much lower -- 1-3 cm/ μ s ($1-3 \times 10^4$ m/s). Therefore, it is desirable that the authors indicate the procedure for measuring the speed: what points on which frames did the authors take to measure the speed. The average quasi-continuous propagation velocity of the leader (at front 100-250 μ s) is about 1-2 cm/ μ s, and this is about 10 times less than the speed of 1.58×10^5 m/s.

Response: The procedure for estimating the speed has been given in the revised manuscript under the guidance of the reviewer. The bi-directional development velocity of the separated luminous structure is indeed significantly larger than the development velocity of the quasi-continuous propagating positive leader. That also confuses us. In our view, one possible explanation may be that the separated luminous structure develops in the leader corona zone, in which the average electric field is about 400-500 kV/m, and the separated luminous structure in fact connects to the main leader tip through a common streamer zone. As a result, the separated luminous structure is indeed significantly larger than the development velocity of the quasi-continuous propagating positive leader.

Comment: Lines 75-77: *Thus, we infer that the space stem/leader lifetime (defined as the time from the emergence of the space stem to the full integration of the space leader into the primary leader channel in this letter) is less than 2.22 μ s in the event.*

This phrase needs to be corrected, as the authors must prove that they see the quasi-space stem.

Response: We have realized the problem. The definition of *the lifetime of space stem/leader* lacked a scientific basis. We have corrected the problem in the revised manuscript. Thanks!

Comment: What do the numbers (10-120, 10-50) mean in color illustrations? It is necessary to set the dimensions in the Figure so that the readers can themselves estimate the velocities and sizes of the plasma formations. For Figure 1, the authors also need to give a current oscillogram, as for Figure 2, as it will greatly complement the information that relates to this unique phenomenon

Response: The coordinate marked in the pseudo-color images represents the row and column numbers of the grayscale matrix. This point was not made clear to readers in the previous manuscript. We have corrected the problem in the revised manuscript.

Unfortunately, due to the failure of the current measurement system, the current waveform of the positive leader discharge was not be obtained. We are sorry!

Comment: Lines 83-84: *The lightning-like discharge is produced by applying the standard positive switching impulse (SI) to a 4 m point-to-plane gap in the summer of 2020.* It is desirable that the authors give the exact value of the front and the duration of the voltage pulse (250/3500 μ s?), Since not all readers of the article work in the field of high-voltage technology.

Response: We have corrected the problem in the **Method** section of the revised manuscript under the guidance of the reviewer. Thanks!

Comment: Lines 84-85: *The absolute humidity is about 20 g/m³, the air pressure is 1 atm.* With what device and with what error did the authors measure humidity? Humidity is a very important parameter for the appearance of positive leader steps.

Response: The humidity measurement instrument and the accuracy has been given in the **Method** section of the revised manuscript under the guidance of the reviewer. Thanks!

Comment: Figure 2. It is desirable that the authors synchronize the frames and the current and give the entire oscillogram of the current and voltage that correspond to this phenomenon. What do the numbers (10-120, 10-50) mean in color drawings? It is necessary to supply the dimensions in the figure. The authors again incorrectly use the term space-stem, since they have no experimental evidence that this plasma formation is precisely a space-stem. Without a clear image of the streamer zone, it is impossible to separate the space stem from the space leader. Most likely a space leader is depicted here. In case of difficulties with the identification of plasma objects, it is reasonable for the authors to write "plasma object" or "plasma formation". The 169 ns figure is printed in the wrong place, as it falls on the dotted line

Response: The synchronized the frames and the current, associated with a positive leader step led by a separate luminous structure, is shown in Figure 3 under the direction of the comment. Unfortunately, only a part of the applied voltage waveform was registered. The coordinate marked in the pseudo-color images represents the row and column numbers of the grayscale matrix. This point was not made clear to readers in the previous manuscript. We have corrected the problem in the revised manuscript. We have replaced the *space stem* with the *separated luminous structure* in the revised manuscript. Thank the reviewer for pointing these problems.

Comment: Lines 93-95: *We infer that the current rise in this stage corresponds to the collision of opposite polarity streamers emerging from the descending leader channel tip and the space stem, respectively.*

What are the authors' grounds for such assumptions and inventing a new term (collision)? The breakthrough phase of interaction between the leaders, with the exception of the first few nanoseconds, is carried out by positive streamers, since the electric field required for the movement of positive streamers is two times less than for negative ones. If the authors do not have evidence of some facts, then it is advisable not to write them. Authors can write their guesses in the discussion of experimental results

Response: We have realized the problem under the direction of the comment. In the revised manuscript, we replace the *collision stage* with the *stage A*. Thanks!

Comment: Lines 96-98: *When the space leader merges with the primary positive leader, the primary positive leader tip's high potential is transferred to the lower end of the space leader, followed by intense positive leader corona streamers burst reported by previous literature.*

Dear colleagues, if you refer to literature, it is always necessary to indicate specific articles and books.

Response: We are sorry for the mistake!

Comment: Lines 100: The third stage is referred to as the decay stage here, lasting several microseconds.

The text does not make it clear to the reader what exactly is disintegrating? If in the experiment the voltage pulse lasted for several ms, then the plasma decay will also last for several ms, and the authors should see the plasma decay in many hundreds of subsequent frames.

Response: The incorrect statement had been deleted in the revised manuscript. Thanks!

Comment: Lines 101-103: Figure 3 shows the current pulse of a negative leader step led by a space stem and 6 consecutive frames recording the step. The space stem emerged at 0.4 m from the descending leader channel tip. The space stem/lifetime is longer than 4.44 μ s.

The authors incorrectly write the term space-stem, since the analysis of Figure 3 does not allow us to establish this. The authors can only correctly assume that this is a space-stem. Also, the authors cannot accurately say the lifetime of the space stem if they cannot accurately establish the nature of the plasma object.

Response: These problems have been corrected in the revised manuscript. Thanks!

Comment: Lines 103-104: The space leader's estimated backward velocity is larger than 6.7×10^5 m/s, which is close to the estimated forward velocity of the space leader shown in Figure 1.

It is desirable that the authors specify exactly what backward velocity is and describe in detail the procedure for measuring the velocity: which points on which frames and what time interval the authors used to measure the velocity. And everywhere it is desirable that the authors indicate that they measure the average speed over a certain period of time.

Response: The procedure for estimating the speed has been given in the revised manuscript under the guidance of the reviewer.

Comment: Lines 106-107: The difference is that the collision stage duration of the negative leader step (lasting about 3.5 μ s) is obviously longer than that of the positive leader step led by the space stem.

The long spark literature does not include such a term as the collision stage. The literature contains such a well-known term as - end-to-end phase. If the authors introduce a new term, then they must have very good reasons, which they must detail in the text.

Response: The problem has been solved in the revised manuscript. Thanks!

Comment: That may indicate that the space stem/leader lifetime in positive leader steps tends to be shorter than in negative leader steps.

This statement is not correct and must be removed, since this conclusion does not follow from experiments

Response: The incorrect statement has been removed. Thank the reviewer for pointing the problem.

Comment: *Lines 131-132. The results reported here powerfully proves the existence of space stem/leader in positive leader steps, ending a long-standing argument.*

It is desirable that the authors change this phrase. The paper produced an important result, shown in Figure 1, but the dispute not only does not end, but only begins. One single experiment cannot serve as definitive proof.

Response: We are sorry for the inappropriate statement, and the problem have been corrected in the revised manuscript.

Comment: *Lines 134-135. The intense leader corona streamers burst occurring after the space leader merges with the primary positive leader injects a strong current pulse into the leader channel.*

This phrase needs to be changed. Figures 2,3 do not allow to distinguish streamer flashes from the powerful illumination of the camera matrix, which occurs during the step. Figure 1 clearly shows that the speed camera does not reliably identify streamers.

Response: Under the direction of the comment, we realized that the statement is not a fact observed in the experiments, just an assumption. We have changed the phrase as “*the primary positive leader would inject a strong current pulse into the leader channel*” in the revised manuscript.

Comment: *Lines 143-144. VHF radiation from the positive lightning leaders is still not well understood. The needle-like discharge that may occur after positive leader development pausing is identified as a critical source.*

It is advisable that the authors make a reference to the paper they are discussing. Need to explain what a critical source is?

Response: We have corrected the problem in the revised manuscript. Thank the reviewer!

Comment: *That indicates the absolute humidity may play an important role in the emergence of the space stem/leader in positive leader steps.*

This conclusion was finally made in 1977 by (Les Renardieres Group (1977)).

Response: We have corrected the problem in the revised manuscript from line 242 to 244. Thank the reviewer!

Comment: *Lines 166-180. Method.* This work is experimental. Therefore, it is desirable that the authors describe in detail the measurement methods, and not give reasoning on general topics. It is desirable that the authors give: the parameters of the bandwidth of the oscilloscope, complete oscillograms of the current (at the high-voltage electrode?) And the

discharge voltage, the parameters of the measuring shunt, the specific brand of the high-speed camera, the sensitivity of the camera matrix depending on the wavelength. The general scheme of the experiment and at what distance the camera was from the discharge would be very useful to the readers. How and with what error were measured humidity, temperature, etc.

Response: Under the guidance of three reviewer, we have rewritten the **Method** section in the revised manuscript. Thank the three reviewers very much!

Comment: *Lines 171-172. The space stem/leader leading to negative leader steps was first discovered in the laboratory³⁵⁻³⁷ and later confirmed through natural lightning observations³⁸.*

This phrase needs reworking. For the first time, the structure of the streamer crown of the negative leader of a long spark was established Stekolnikov and Shkilev, (1963). The article by Mitchell and Snoddy (1947) has nothing to do with a long spark and should be deleted as the experiments were carried out in tubes (not free space) at very low pressure and low voltage.

Response: The incorrect statement has been deleted in the revised manuscript. The comment is constructive and important for us! Thanks!

Thank the reviewer very much!

REVIEWER COMMENTS

Reviewer #1 (Remarks to the Author):

Review Attached

In this revision the authors have significantly improved their work, now titled “Separated luminous structures leading positive leader steps”. However, I still have significant concerns regarding the authors’ interpretation of their data. I will address my concerns roughly in order they appear in the paper.

In the abstract and introduction the authors state:

The polarity asymmetry of leaders describes the difference in macroscopic behavior between positive and negative leaders and is a long-term consensus among lightning physicists.

This sentence needs improvement. As it begs the question of what differences does “polarity asymmetry” refer to. I recommend a sentence along the lines of “It is known that positive and negative leaders seem to behave very differently. For example, negative leaders are known to step where it is not known if positive leaders do (exact example doesn’t matter), these differences are collectively referred to as ‘polarity asymmetry’ between positive and negative leaders”.

On lines 50-51 the authors claim

The VHF radiation emitted from positive leader bodies is attributed to the needle-like discharge recently discovered.

However, recoil leaders can also emit VHF from positive leaders (see many references in ref 20, and elsewhere), and thus should be mentioned.

On lines 151, 153, 155, and elsewhere, the authors use the high-speed camera footage to estimate speed of propagation. However, these phenomena are only seen on a few camera frames, which means that the duration of each camera frame will result in large errors in velocity (i.e. did the propagation start at the beginning or end of the frame?). Thus the authors need to precisely detail how they calculated these speeds (particularly the durations), and give a sense of the error. Since the error bars must be quite large, I recommend that the authors give a range of speeds that arrive from different assumptions (such as if propagation starts at beginning or end of a frame).

On lines 127-128, and 165-171 the authors claim that one corona region is stronger than another. I think this is an extremely ambiguous claim that should be removed. First is the definition of “stronger”. One corona region is possibly more luminous, but streamers are not in thermal equilibrium thus it is not at all clear why one region is brighter than the other. Thus, the word “stronger” here is far too vague. In addition, it isn’t even clear to me that one region is brighter. This all depends on which pixels you assume are corona or leader. There are parts of the leader that are quite dim, so it seems entirely impossible to me to say that a particularly bright pixel is definitely part of the corona and not a leader. Therefore, the portions of the lower-corona that more bright may not be corona at all, but may be a space leader. Thus, I’d argue it is entirely unclear from this data what is happening on this scale.

In Lines 172 – 184 the authors present a hypothesis about what is happening in their data. However, while I particularly like this hypothesis, I don’t see that it is supported by this data. Primarily due to

the issues I stated above. Note that luminosity says nothing about about state of plasma. It only relates to temperature and conductivity if in thermal equilibrium, which streamers are not. Thus, it is entirely impossible to relate this data to physical parameters like temperature and conductivity. Similarly, the authors make many arguments about potential, but it is not at all clear to me what the connection between 2D luminosity and electric potential is. Most of the leader must be at the same potential, but the brightness of a leader along a channel varies drastically.

In Line 96, Fig 2, and line 187, the authors claim that figure 2 shows a corona burst. While I believe a corona burst must be there, I see no evidence of it in figure 2. Frame 5, in particular, shows the channel getting extremely bright, which could be due to multiple reasons, of which a corona burst is only one. To show a corona burst you need to show evidence of many streamers propagating forward very suddenly. I do not see any of this in frame 5 of figure 2. I only see a sudden increase in brightness, and perhaps a small increase in channel length, which could be due to many reasons (like the luminous structure simply continuing to propagate forward without a corona burst).

In lines 187-190 the authors discuss this corona burst perhaps injecting a strong current. Again, while this is a hypothesis I am very favorable towards, I see no evidence for it in this data set.

Fig 5 shows current and high-speed video frames. However there is no indication on how the current lines up with the high-speed video (as done in figure 3). The authors should do something similar to figure 3 to show how the current lines up with video.

Line 238 should reference "VHF and UHF Electromagnetic Radiation Produced by Streamers in Lightning" by Shi, Liu, et al. in 2019.

Reviewer #2 (Remarks to the Author):

In this revision of the manuscript the authors have addressed my previous criticisms and I believe that the paper is substantially improved. The methods section gives now many more details about the experimental setup.

However I still found some minor issues that I would ask the authors to address before the final version can be published:

- Line 64: I think that the time resolution of a camera usually refers to the framerate. So although in some cases the exposure times used in this work are below one microsecond, I would not speak about sub-microsecond time resolution.

- Line 85: This is also a minor issue but I do not think that using false colors to improve the visualization of an image counts as "processing".

- Line 88: Gary -> grayscale

- Lines 124-128: It is unclear what a "stronger" discharge means. Is it a higher current, more streamers, more dissipated energy? The authors should be careful to conclude any of these things just from a higher luminance, although it may be a hint.

- Line 169 and others: "like-common-streamer-zone" is a awkward name. Perhaps "streamer-like common zone"?

- Line 201: "very" -> "every".

- Line 202: Please clarify here "positive space leader", which I believe is the intended meaning of the sentence.

- Line 230: I did not find a precise definition of the "stage A" in the current profile. Looking at figure 5 the starting point of this stage seems to be quite arbitrary. A proper comparison between the duration of these stages in positive and negative leaders requires a precise definition.

- Line 281: Remove repeated words.

Reviewer #3 (Remarks to the Author):

The reviewer uploaded the review as a pdf file

Review 2

The paper, in comparison with the first version, has a new, very important result. For the first time, for a long positive spark, the authors were able to obtain an image of a “separate luminous structure” (most likely a quasi-space leader) together with a clearly visible positive streamer corona located between the main channel of the long positive spark and the space leader. It is very important that the authors obtained these images simultaneously with the recording of the electric current coming from the high-voltage electrode. [Kostinskiy et al., 2018] for the first time obtained images of a separate luminous structure, but in their studies, it was impossible to reliably identify the streamer corona between the separate luminous structure and the main channel of a long positive spark, and [Kostinskiy et al., 2018] did not measure simultaneously the current and the images a separate luminous structure.

The paper as a whole is much better than in the first version, but unfortunately it still needs corrections and improvements and therefore the paper cannot be published in the current version.

It is advisable that the authors use a spell checker, as the paper has a lot of spelling errors.

Line 27. The physics governing the propagation of lightning leaders are still not well understood.

The authors study a long spark, not lightning, and they often unreasonably transfer the results obtained for a long spark to lightning. At present, the identity of the physical processes in the streamer zone of lightning and a long spark has not been proven. Today we can only talk about the similarity of processes, but not their identity. Therefore, it is advisable that the authors correct this proposal, for example, like this: “The physics governing the propagation of lightning leaders and long spark leaders are still not well understood”.

Line 29-31. It is generally believed that negative leader steps are led by the separated luminous structures termed space stems/ leaders and there is no space stem/leader in positive leader discharges.

The authors of the paper are studying a long spark. Classic works [Stekolnikov and Shkilyov, 1963; Les Renardieres Group, 1981; Gorin et al., 1976; Bazelyann and Raizer 1998; Rakov and Uman, 2003; Kostinskiy et al., 2018] clearly established the structure of the streamer zone of the negative long spark (see the Figure below). The space stems are at the front of the streamer zone and “lead” the space leaders, the space leaders follow the space stems and, in turn, “lead” the main negative leader channel, which moves from the high voltage electrode. It is the space stems that “lead” the entire streamer zone and the negative leader, and the steps are formed during the connection of the space leader and the main negative leader. Therefore, in order to correspond to the physics of phenomena, which is reliably established, the authors should rewrite this proposal, perhaps as follows: “It is generally believed that negative leader steps are led by the separated luminous structures termed space stems and space leaders and there is no separated luminous structures in positive leader discharges”.

Line 31-33. A separated luminous structure had been recently observed in positive leader discharges [Kostinskiy et al., 2018], suggesting that positive leaders may do steps in a similar way to negative leaders under certain conditions.

It is desirable for the authors to supplement this sentence with a reference to the article in which this discovery was made, so that the paper does not create the false impression that the authors of this paper were the first to make this observation. The reviewer has underlined the desired addition in the text.

Downward Negative Leader Stepping Mechanism

Gorin et al. [1978]

Line 34. However, there is no direct evidence to support this hypothesis.

The observation of a separate luminous structure in the article [Kostinskiy et al., 2018] is direct evidence of the existence of this separate plasma formation, the same direct evidence as the images obtained in the paper presented by the authors.

Line 38-39. We hope these findings would deepen the understanding of the nature underlying positive long spark leaders and lightning leaders

The authors of the paper study a long spark, not lightning, and their main results relate to a long spark, which, perhaps, has a high similarity to lightning (this fact is currently being proved, not proven). The reviewer has underlined the desired addition in the text.

Line 45-46. Negative leaders propagate in discrete steps, led by the separated luminous structures called space stems and space leaders, with rapid elongations and sharp channel illuminations³⁻¹⁰.

These lines need correction (see Notes Line 29-31). The reviewer has underlined the desired addition in the text.

Line 45-46. Recently, for the first time, a separated luminous structure had been observed inside the leader corona zone in positive leader discharges²⁷.

These lines need correction. The reviewer has underlined the desired addition in the text.

Line 45-46. The nature of the steps/restrikes occurring during the quasi-continuous positive leader development at high humidity is still in doubt²⁸.

This reference to the article is incorrect, since the authors of [28] investigate and discuss the flash mode [Bazelyan & Raizer, 1998, pp. 205] of leader propagation near the high-voltage electrode, when the voltage is applied to the electrode very slowly (less than 0.5 kV/ μ s). Article [28] does not discuss the role of humidity and steps that arise during the quasi-continuous phase of the leader's development.

Line 65-67. *Our result suggests that positive leaders could propagate in a similar manner as negative leaders at high humidity, which deepens understanding of the physical mechanisms governing the propagation of the positive leader.*

There can be no complete analogy of the mechanism of steps between the negative and positive leaders, since the entire streamer zone of the negative leader is led by space stems, and in the positive leader the existence of space stems was not found (including this paper).

Therefore, the authors could more accurately write something like this: “Our result suggests that, at least at high absolute humidity, positive leaders can form steps due to the merging of a separate plasma formation and the main channel of a positive leader, similar to the steps of a negative leader, which are formed due to merging the space leader with the main negative leader”.

Line 76-77. *A diffuse luminous area, morphologically similar to the common streamer zone in the breakthrough phase²⁹, connects the upper end of the separated luminous **structure** to the primary leader channel.*

The reference to paper [29] is incorrect, since the experimental data of [29] do not allow one to detect the streamer zone between the leader channels. Currently, in lightning research, there are no experiments with high-speed cameras that would reliably fix streamers in the streamer zone of lightning leaders, since the matrix of high-speed cameras does not allow this to be done at distances greater than 500-700 meters.

Line 83. It is desirable that the captions under all the pictures have the name of the high-speed camera with which they were obtained, since high-speed cameras had different matrices and a different number of pixels for each frame.

Line 89-90. *The **color (colour?)** of pixels in these pseudo-**color** images represent these value in the grayscale matrix. The deepest red is corresponding to the maximum grayscale value of 2460.*

What are the luminance values of the channel in Panel # 5 if they are white? These pixels must be in a saturation state, i.e. be brighter than the reddest pixels. It is desirable that, out of respect for the editorial board and reviewers, the authors use a spell checker.

Line 96-99. *Comparing the morphology of the leader channel recorded by frame #5 and the separated luminous **structure**, it is suggested that the separated luminous **structure** became the newly added section of the leader channel in the step, causing the abrupt elongation of the descending leader channel.*

Perhaps the authors should supplement the proof of the existence of a step with several additional important arguments. The step of the positive leader is almost a straight line (Figure 1 # 7), as in the studies [Kostinskiy et al, 2018, Figure 9-12], which is fundamentally different from the tortuous trajectory of the channel development during the quasi-continuous movement of the positive leader of a long spark. The step current of the positive leader rises sharply by an order of magnitude upon initiation of the step (Figure 3), as in the studies [Kostinskiy et al, 2018, Figure 12c]. Negative leader steps are also always accompanied by a sharp increase in current (Figure 5), which is a well-established fact, both for a long spark and for lightning.

Line 150-151. *The estimated backward velocity of the separated luminous structure is about 1.56×10^5 m/s.*

Since the authors everywhere estimate the velocity using the time between frames plus the exposure time of the frame, they everywhere measure the average velocity for this period of time. Therefore, it is desirable that the authors write the term “average velocity for this period of time” throughout the text and necessarily calculate the error in measuring the speed.

Line 156-157. *The length of the separated luminous structure recorded in frame #8 is about 1 m.*

It is desirable that the authors add something like this: “This length is close in size previously measured [Les Renardieres Group, 1977; Kostinskiy et al, 2018]”, since the length of the steps of a positive leader was measured in earlier works.

Line 164. *Potentail* transfer and current pulse

Authors should probably write the word "Potential" or explain whose tail it is ☹️

Line 165-171. *According to the result shown in Figure 2, it is suggested that the corona streamer discharge emerging from the bottom end of the separated luminous structure in the exposure duration of frame #4 is stronger than the corona streamer discharge emerging from the descending leader tip in the leader quasi-continuous development phase. Meanwhile, the upper end of the separated luminous structure had been connected to the primary leader channel through a like-common-streamer-zone. We infer that the stronger corona streamer discharge is due to that the high potential of the primary leader tip has been partly transferred to the bottom end of the separated luminous structure.*

This conclusion looks strange, it most likely needs to be removed or fundamentally changed. At the same potentials at the electrode, the potential of the head of the positive leader in the quasi-continuous phase is greater than the potential at the lower head of a separate luminous structure, since a part of the total potential of the electrode drops (is lost) in the streamer zone and channel of a separate luminous structure.

Line 198-206. *Comparision* of leader steps of different polarities. *5 consecutive frames recording a typical negative leader step are shown in Figure 5. The negative leader discharge was produced in a rod-to-plane air gap with 2 m length on June 22nd, 2020. The bi-directional development of the space leader was imaged by 3 consecutive frames (#1, #2 and #3) in the event and the processes were recorded preceding almost very negative leader steps in our research, while there is only one event that the bi-directional development of the space leader was imaged by 2 consecutive frames. That indicates that the bi-directional development time of the separated luminous structure leading positive leader steps may be shorter than that of space leaders leading negative leader steps. This may explain why the space leader leading negative leader steps are easily discovered than the separated luminous structure leading positive leader steps, which require higher temporal resolution for observation.*

This section describes the original experimental data, so the authors are encouraged to remove the experimental data from the "Discussion" section and move them to the "Results" section. The authors unsuccessfully chose the example of a negative leader step, since in frames # 4.5 (Fig. 5) we actually see 2 steps, and this makes it difficult to compare with one step of a positive leader. It is desirable that the authors find a clearer example of a single negative leader stage.

In Figure 5, the authors did not very clearly indicate the synchronization of the exposure time of the frames with the current oscillogram. It is desirable that the authors place the exposure time of the frames in the same way as in Figure 3. Frames # 1, # 2, # 3 show only a part of the space leader and one cannot see the bidirectional development of the space leader on them. It is advisable to delete the last two sentences, since such strong statements cannot be made based on only three events. For such strong conclusions, it is desirable to carry out hundreds of measurements of steps of each polarity or refer to articles where these results were obtained earlier.

Line 217-219. *Comparing the morphology of the space leader shown in frames #1 and #3 and considering the frame interval is 1.11 μ s, the estimated backward velocity of the space leader is about 1.17×10^5 m/s, and the estimated forward velocity of the descending negative leader is about 0.63×10^5 m/s during the phase.*

All velocities are average velocities and the authors would like to estimate the measurement errors of all velocities (see lines 150-151).

Line 219-224. *The estimated forward velocity of the descending leader is smaller than the estimated backward velocity of the space leader during the connection of the space leader to the primary leader channel, which is different from the positive leader steps shown in Figure 4. We infer that the smaller forward velocity of the descending primary leader may be the reason for the longer duration of the bi-directional development of the space leader in negative leader steps. Note that a common streamer zone could be seen in frame #2.*

It is advisable to delete this fragment, since you cannot make such strong statements based on just three events. For such strong conclusions, it is desirable to carry out hundreds of measurements of steps of each polarity. Moreover, these are very rough estimates in comparison with the classical works [Gorin et al. 1976, Les Renardieres Group, 1977], in which measurements were made using continuous scanning of images and continuous measurement of velocities. The average speed significantly distorts the physics of the breakthrough phase, where the speed of the leaders' movement changes rapidly and is not constant.

Line 242-263. Emergence of the separated luminous structure in positive leader discharge

It is advisable to remove this section from the paper, since the authors wrote it so badly that it cannot be reviewed, since there are mistakes and misunderstandings in each sentence. Unfortunately, the review genre does not allow the reviewer to turn it into a lecture on plasma physics and long spark kinetics. The authors obtained an important experimental result, and this result does not weaken, the fact that the authors do not provide an adequate explanation, since at the moment no one has an adequate explanation of this phenomenon.

Line 265-266. *The results presented above indicate that separated luminous structures in positive leader discharges could lead steps at high humidity, like that the space leader leads negative leader steps.*

It is reasonable for the authors to rewrite these lines, since the negative leader of a long spark in the discharge gaps of more than three meters always moves in steps, regardless of humidity.

Line 274-275. *All these findings indicate that positive leaders could do steps in a similar way to negative leaders at high humidity.*

It is reasonable for the authors to rewrite these lines, since the negative leader of a long spark in the discharge gaps of more than three meters always moves in steps, regardless of humidity.

Line 275-276. *One problem is solved, but more interesting questions are raised.*

It is desirable to remove this sentence, since the problem cannot be solved on the basis of two observations. The study of the fine structure of the steps of a positive leader is just beginning.

Line 281-282. *All records presented **in this paper** were obtained at the Ultra-High-Voltage **in this paper** were obtained at the Ultra-High Voltage Laboratory in Hefei, China.*

Line 280-281. Method.

It is desirable to move the "Method" section to the beginning of the paper.

The information presented in this section needs to be supplemented:

- On all Figures it is desirable to give an oscillogram of voltage measurements.

- Where is the Marx generator located indoors or outdoors?
- How far from the tip of the HV-electrode were the high-speed cameras?
- What lenses were used for speed cameras?
- Have trigger delays been measured to accurately synchronize current, voltage, and exposure measurements on speed cameras?

References

Bazelyan, E. M., & Raizer, Y. P. (1998). Spark discharge (p. 294). Boca Raton, FL: CRC Press.

Gorin, B. N., Levitov, V. I., & Shkilev, A. V. (1976). Some principles of leader discharge of air gaps with a strong non-uniform field. IEE Conference Publication, 143, 274–278.

Kochkin, P O, C V Nguyen, A P J van Deursen and U Ebert (2012), Experimental study of hard X-rays emitted from meter-scale positive discharges in air, arXiv:1208.5899v2 [physics.plasm-ph] 13 Sep 2012

Kostinskiy, A. Y., Syssoev, V. S., Bogatov, N. A., Mareev, E. A., Andreev, M. G., Bulatov, M. U., et al. (2018). Abrupt elongation (stepping) of negative and positive leaders culminating in an intense corona streamer burst: Observations in long sparks and implications for lightning. *Journal of Geophysical Research: Atmospheres*, 123. <https://doi.org/10.1029/2017JD027997>

Les Renardieres Group (1977). Positive discharges in long air gaps at Les Renardieres, 1975 results and conclusions. *Electra*, 53, 31–153.

Rakov, V. A., & Uman, M. A. (2003). *Lightning: Physics and effects* (p. 687). New York: Cambridge University Press. <https://doi.org/10.1017/CBO9781107340886>

Popov, N. A. (2009). Study of the formation and propagation of a leader channel in air. *Plasma Physics Reports*, 35(9), 785–793. <https://doi.org/10.1134/S1063780X09090074>

Stekolnikov I.S. and Shkilyov A.V. (1963). The development a long spark and lightning. Proc. Of the Third I.C. of Atmosphere and Space electricity. Montreux, Switzerland, may, 5-10, pp.466-481

Response to Reviewers

We would first like to express our gratitude to the three reviewers again! They contribute a lot to the birth of a rigorous scientific paper. Every comment is valuable! We learned a lot in the process of revising the manuscript. Due to the COVID-19 and unexpected health problems, we were not able to complete the revision on time. We are sorry for this! We have revised the manuscript accordingly. Our responses are given in a point-by-point manner below.

Response to Reviewer #1

Comment: In the abstract and introduction the authors state: *The polarity asymmetry of leaders describes the difference in macroscopic behavior between positive and negative leaders and is a long-term consensus among lightning physicists.*

This sentence needs improvement. As it begs the question of what differences does “polarity asymmetry” refer to. I recommend a sentence along the lines of “It is known that positive and negative leaders seem to behave very differently. For example, negative leaders are known to step where it is not known if positive leaders do (exact example doesn’t matter), these differences are collectively referred to as ‘polarity asymmetry’ between positive and negative leaders”.

Response: Under the direction, we realize that the statement is inappropriate. We have corrected the problem. Please see lines 28-31.

Comment: On lines 50-51 the authors claim: *The VHF radiation emitted from positive leader bodies is attributed to the needle-like discharge recently discovered.*

However, recoil leaders can also emit VHF from positive leaders (see many references in ref 20, and elsewhere), and thus should be mentioned

Response: The comment is very helpful to us! By reading the references in ref 20 in depth, we realized the problems and made corrections. Please see line 51. Thank you very much!

Comment: On lines 151, 153, 155, and elsewhere, the authors use the high-speed camera footage to estimate speed of propagation. However, these phenomena are only seen on a few camera frames, which means that the duration of each camera frame will result in large errors in velocity (i.e. did the propagation start at the beginning or end of the frame?). Thus the authors need to precisely detail how they calculated these speeds (particularly the durations), and give a sense of the error. Since the error bars must be quite large, I recommend that the authors give a range of speeds that arrive from different assumptions (such as if propagation starts at beginning or end of a frame).

Response: We have made corrections under the guidance of the reviewer. Please see lines 187-193. Thank the reviewer very much!

Comment: On lines 127-128, and 165-171 the authors claim that one corona region is stronger than another. I think this is an extremely ambiguous claim that should be removed. First is the

definition of “stronger”. One corona region is possibly more luminous, but streamers are not in thermal equilibrium thus it is not at all clear why one region is brighter than the other. Thus, the word “stronger” here is far too vague. In addition, it isn’t even clear to me that one region is brighter. This all depends on which pixels you assume are corona or leader. There are parts of the leader that are quite dim, so it seems entirely impossible to me to say that a particularly bright pixel is definitely part of the corona and not a leader. Therefore, the portions of the lower-corona that more bright may not be corona at all, but may be a space leader. Thus, I’d argue it is entirely unclear from this data what is happening on this scale.

Response: Thank the reviewer for pointing the problem! we have realized the problem and removed the inappropriate statement in the revised manuscript.

Comment: In Lines 172 – 184 the authors present a hypothesis about what is happening in their data. However, while I particularly like this hypothesis, I don’t see that it is supported by this data. Primarily due to the issues I stated above. Note that luminosity says nothing about about state of plasma. It only relates to temperature and conductivity if in thermal equilibrium, which streamers are not. Thus, it is entirely impossible to relate this data to physical parameters like temperature and conductivity. Similarly, the authors make many arguments about potential, but it is not at all clear to me what the connection between 2D luminosity and electric potential is. Most of the leader must be at the same potential, but the brightness of a leader along a channel varies drastically.

Response: Under the guidance of the reviewers, we recognize that the data reported in this paper are insufficient to support this hypothesis. In the revised manuscript, we have removed the inappropriate statement. Many thanks to the reviewer!

Comment: In Line 96, Fig 2, and line 187, the authors claim that figure 2 shows a corona burst. While I believe a corona burst must be there, I see no evidence of it in figure 2. Frame 5, in particular, shows the channel getting extremely bright, which could be due to multiple reasons, of which a corona burst is only one. To show a corona burst you need to show evidence of many streamers propagating forward very suddenly. I do not see any of this in frame 5 of figure 2. I only see a sudden increase in brightness, and perhaps a small increase in channel length, which could be due to many reasons (like the luminous structure simply continuing to propagate forward without a corona burst).

Response: Thank the review for pointing this key point. The intensive corona streamer burst was not observed in our experiments. We have revised the statement under the direction of the reviewer. See lines 246-258.

Comment: In lines 187-190 the authors discuss this corona burst perhaps injecting a strong current. Again, while this is a hypothesis I am very favorable towards, I see no evidence for it in this data set.

Response: we realized that the data reported in this paper are insufficient to support this hypothesis. In the revised manuscript, we have removed inappropriate statement.

Comment: Fig 5 shows current and high-speed video frames. However there is no indication on how the current lines up with the high-speed video (as done in figure 3). The authors should

do something similar to figure 3 to show how the current lines up with video.

Response: The problem has been solved in the revised manuscript. Please see Figure 6. Thanks!

Comment: Line 238 should reference “VHF and UHF Electromagnetic Radiation Produced by Streamers in Lightning” by Shi, Liu, et al. in 2019.

Response: Corrected! Please see line 267. Thank you very much!

Thank the reviewer very much!

Response to Reviewer #2

Comment: - Line 64: I think that the time resolution of a camera usually refers to the framerate. So although in some cases the exposure times used in this work are below one microsecond, I would not speak about sub-microsecond time resolution.

Response: The inappropriate statement has been removed in the revised manuscript. Thank you very much!

Comment: - Line 85: This is also a minor issue but I do not think that using false colors to improve the visualization of an image counts as "processing".

Response: Yes! The possibly misleading “unprocessed” has been replaced with “original” in the revised manuscript. Please see the captions of Figure 1 and Figure 3.

Comment: - Line 88: Gary -> grayscale

Response: Corrected! Thank you very much!

Comment: - Lines 124-128: It is unclear what a "stronger" discharge means. Is it a higher current, more streamers, more dissipated energy? The authors should be careful to conclude any of these things just from a higher luminance, although it may be a hint.

Response: The comment is helpful to us! We realized the problem. We have removed the inappropriate statement in the revised manuscript. Thanks!

Comment: - Line 169 and others: "like-common-streamer-zone" is a awkward name. Perhaps "streamer-like common zone"?

Response: In the revised manuscript, "like-common-streamer-zone" has been replaced with “dense luminous spindle”, according to its morphology. Thank you for pointing the problem.

Comment: Line 201: "very" -> "every".

Response: The problem have been solved! Thanks!

Comment: - Line 202: Please clarify here "positive space leader", which I believe is the intended meaning of the sentence.

Response: The problem have been solved!

Comment: - Line 230: I did not find a precise definition of the "stage A" in the current profile. Looking at figure 5 the starting point of this stage seems to be quite arbitrary. A proper comparison between the duration of these stages in positive and negative leaders requires a precise definition.

Response: The comment is very important! As pointed by reviewer #3, we failed to choose the example of a negative leader step, since in frames there are 2 steps, and this makes it difficult to compare with one step of a positive leader. Furthermore, Due to the complexity of negative leader steps and limited by our experiment data, we were not able to give a precise definition of stage A. We are sorry! As a remedy, we have reselected the example of a negative leader step in the revised manuscript. Thank you for the comment very much!

Comment: - Line 281: Remove repeated words.

Response: Corrected! Thank you very much!

Thank the reviewer very much!

Response to Reviewer #3

Comment: It is advisable that the authors use a spell checker, as the paper has a lot of spelling errors.

Response: We are Sorry! We are ashamed of our mistakes! In the revised manuscript, we did our best to avoid spelling mistakes! Thank you for the comment!

Comment: *Line 27. The physics governing the propagation of lightning leaders are still not well understood.*

The authors study a long spark, not lightning, and they often unreasonably transfer the results obtained for a long spark to lightning. At present, the identity of the physical processes in the streamer zone of lightning and a long spark has not been proven. Today we can only talk about the similarity of processes, but not their identity. Therefore, it is advisable that the authors correct this proposal, for example, like this: "The physics governing the propagation of lightning leaders and long spark leaders are still not well understood".

Response: We realized the problem! The statement was also revised according to the reviewer's instructions! Please see line 27. Thank you very much!

Comment: *Line 29-31. It is generally believed that negative leader steps are led by the separated luminous structures termed space stems/ leaders and there is no space stem/leader in positive leader discharges.*

The authors of the paper are studying a long spark. Classic works [Stekolnikov and Shkilyov, 1963; Les Renardieres Group, 1981; Gorin et al., 1976; Bazelyann and Raizer 1998; Rakov and Uman, 2003; Kostinskiy et al., 2018] clearly established the structure of the streamer zone of the negative long spark (see the Figure below). The space stems are at the front of the streamer zone and "lead" the space leaders, the space leaders follow the space stems and, in turn, "lead" the main negative leader channel, which moves from the high voltage electrode. It is the space stems that "lead" the entire streamer zone and the negative leader, and the steps are formed

during the connection of the space leader and the main negative leader. Therefore, in order to correspond to the physics of phenomena, which is reliably established, the authors should rewrite this proposal, perhaps as follows: “It is generally believed that negative leader steps are led by the separated luminous structures termed space stems and space leaders and there is no separated luminous structures in positive leader discharges”.

Response: Thank the reviewer for the instructions very much! We have revised the statement according to the reviewer’ s instructions! Please see lines 28-31. Thank you very much!

Comment: *Line 31-33. A separated luminous structure had been recently observed in positive leader discharges [Kostinskiy et al., 2018], suggesting that positive leaders may do steps in a similar way to negative leaders under certain conditions.*

It is desirable for the authors to supplement this sentence with a reference to the article in which this discovery was made, so that the paper does not create the false impression that the authors of this paper were the first to make this observation. The reviewer has underlined the desired addition in the text.

Response: The problem is serious! However, we were not sure if references could be allowed to add to the abstract, and as a remedy, we modified the expression as “*a separate luminous structure observed in a positive leader discharge had been reported in recent literature, suggesting that positive leaders may similarly do steps to negative leaders under certain conditions.*”. Thank you for pointing the problem.

Comment: *Line 34. However, there is no direct evidence to support this hypothesis.*

The observation of a separate luminous structure in the article [Kostinskiy et al., 2018] is direct evidence of the existence of this separate plasma formation, the same direct evidence as the images obtained in the paper presented by the authors.

Response: Corrected! Please see line 33. Thanks!

Comment: *Line 38-39. We hope these findings would deepen the understanding of the nature underlying positive long spark leaders and lightning leaders.*

The authors of the paper study a long spark, not lightning, and their main results relate to a long spark, which, perhaps, has a high similarity to lightning (this fact is currently being proved, not proven). The reviewer has underlined the desired addition in the text.

Response: We have revised the statement according to the reviewer’ s instructions! Please see lines 37-28. Thank you very much!

Comment: *Line 45-46. Negative leaders propagate in discrete steps, led by the separated luminous structures called space stems and space leaders, with rapid elongations and sharp channel illuminations.*

These lines need correction (see Notes Line 29-31). The reviewer has underlined the desired addition in the text.

Response: We have revised the statement according to the reviewer’ s instructions! Please see lines 45-46. Thanks!

Comment: *Line 45-46. Recently, for the first time, a separated luminous structure had been*

observed inside the leader corona zone in positive leader discharges.

These lines need correction. The reviewer has underlined the desired addition in the text.

Response: We have made corrections according to the reviewer's instructions! Please see line 57. Thanks!

Comment: *Line 45-46. The nature of the steps/restrikes occurring during the quasi-continuous positive leader development at high humidity is still in doubt²⁸.*

This reference to the article is incorrect, since the authors of [28] investigate and discuss the flash mode [Bazelyan & Raizer, 1998, pp. 205] of leader propagation near the high-voltage electrode, when the voltage is applied to the electrode very slowly (less than 0.5 kV/ μ s). Article [28] does not discuss the role of humidity and steps that arise during the quasi-continuous phase of the leader's development.

Response: Thank you for pointing the problem! we have made corrections. Please see line 62.

Comment: *Line 65-67. Our result suggests that positive leaders could propagate in a similar manner as negative leaders at high humidity, which deepens understanding of the physical mechanisms governing the propagation of the positive leader.*

There can be no complete analogy of the mechanism of steps between the negative and positive leaders, since the entire streamer zone of the negative leader is led by space stems, and in the positive leader the existence of space stems was not found (including this paper). Therefore, the authors could more accurately write something like this: "Our result suggests that, at least at high absolute humidity, positive leaders can form steps due to the merging of a separate plasma formation and the main channel of a positive leader, similar to the steps of a negative leader, which are formed due to merging the space leader with the main negative leader".

Response: Thank the reviewer for the instructions! We have revised the statement. Please see lines 67-69.

Comment: *"Line 76-77. A diffuse luminous area, morphologically similar to the common streamer zone in the breakthrough phase²⁹, connects the upper end of the separated luminous structure to the primary leader channel.*

The reference to paper [29] is incorrect, since the experimental data of [29] do not allow one to detect the streamer zone between the leader channels. Currently, in lightning research, there are no experiments with high-speed cameras that would reliably fix streamers in the streamer zone of lightning leaders, since the matrix of high-speed cameras does not allow this to be done at distances greater than 500-700 meters.

Response: Corrected! Please see line 152. Thank you very much!

Comment: Line 83. It is desirable that the captions under all the pictures have the name of the high-speed camera with which they were obtained, since high-speed cameras had different matrices and a different number of pixels for each frame.

Response: We have revised the manuscript under the direction. Please see the captions under all the pictures. Thank you!

Comment: The color (colour?) of pixels in these pseudo-color images represent these value in

the grayscale matrix. The deepest red is corresponding to the maximum grayscale value of 2460. What are the luminance values of the channel in Panel # 5 if they are white? These pixels must be in a saturation state, i.e. be brighter than the reddest pixels. It is desirable that, out of respect for the editorial board and reviewers, the authors use a spell checker.

Response: Yes! The pixels showing white are in a saturation state. In the revised draft, we have explained this. We must apologize for our disastrous spelling mistake. Sorry!

Comment: Line 96-99. Comparing the morphology of the leader channel recorded by frame #5 and the separated luminous structure, it is suggested that the separated luminous structure became the newly added section of the leader channel in the step, causing the abrupt elongation of the descending leader channel.

Perhaps the authors should supplement the proof of the existence of a step with several additional important arguments. The step of the positive leader is almost a straight line (Figure 1 # 7), as in the studies [Kostinskiy et al, 2018, Figure 9-12], which is fundamentally different from the tortuous trajectory of the channel development during the quasi-continuous movement of the positive leader of a long spark. The step current of the positive leader rises sharply by an order of magnitude upon initiation of the step (Figure 3), as in the studies [Kostinskiy et al, 2018, Figure 12c]. Negative leader steps are also always accompanied by a sharp increase in current (Figure 5), which is a well-established fact, both for a long spark and for lightning.

Response: Thank the reviewer for the instructions! We have revised the statement. Please see lines 84 and 107. Thank you very much!

Comment: Line 150-151. The estimated backward velocity of the separated luminous structure is about 1.56×10^5 m/s.

Since the authors everywhere estimate the velocity using the time between frames plus the exposure time of the frame, they everywhere measure the average velocity for this period of time. Therefore, it is desirable that the authors write the term “average velocity for this period of time” throughout the text and necessarily calculate the error in measuring the speed.

Response: Corrected! Thank you and reviewer #1.

Comment: Line 156-157. The length of the separated luminous structure recorded in frame #8 is about 1 m.

It is desirable that the authors add something like this: “This length is close in size previously measured [Les Renardieres Group, 1977; Kostinskiy et al, 2018]”, since the length of the steps of a positive leader was measured in earlier works.

Response: We have revised the manuscript. Please see line 186.

Comment: Line 164.

Response: We are sorry for the mistake !

Comment: Line 165-171. According to the result shown in Figure 2, it is suggested that the corona streamer discharge emerging from the bottom end of the separated luminous structure in the exposure duration of frame #4 is stronger than the corona streamer discharge emerging from the descending leader tip in the leader quasi-continuous development phase. Meanwhile,

the upper end of the separated luminous structure had been connected to the primary leader channel through a like-common-streamer-zone. We infer that the stronger corona streamer discharge is due to that the high potential of the primary leader tip has been partly transferred to the bottom end of the separated luminous structure.

This conclusion looks strange, it most likely needs to be removed or fundamentally changed. At the same potentials at the electrode, the potential of the head of the positive leader in the quasi-continuous phase is greater than the potential at the lower head of a separate luminous structure, since a part of the total potential of the electrode drops (is lost) in the streamer zone and channel of a separate luminous structure.

Response: Thank you and the reviewer #1! This inappropriate statement has been removed from the revised draft!

Comment: *Line 198-206. Comparison of leader steps of different polarities. 5 consecutive frames recording a typical negative leader step are shown in Figure 5. The negative leader discharge was produced in a rod-to-plane air gap with 2 m length on June 22nd, 2020. The bi-directional development of the space leader was imaged by 3 consecutive frames (#1, #2 and #3) in the event and the processes were recorded preceding almost very negative leader steps in our research, while there is only one event that the bi-directional development of the space leader was imaged by 2 consecutive frames. That indicates that the bi-directional development time of the separated luminous structure leading positive leader steps may be shorter than that of space leaders leading negative leader steps. This may explain why the space leader leading negative leader steps are easily discovered than the separated luminous structure leading positive leader steps, which require higher temporal resolution for observation.*

This section describes the original experimental data, so the authors are encouraged to remove the experimental data from the "Discussion" section and move them to the "Results" section. The authors unsuccessfully chose the example of a negative leader step, since in frames # 4.5 (Fig. 5) we actually see 2 steps, and this makes it difficult to compare with one step of a positive leader. It is desirable that the authors find a clearer example of a single negative leader stage. In Figure 5, the authors did not very clearly indicate the synchronization of the exposure time of the frames with the current oscillogram. It is desirable that the authors place the exposure time of the frames in the same way as in Figure 3. Frames # 1, # 2, # 3 show only a part of the space leader and one cannot see the bidirectional development of the space leader on them. It is advisable to delete the last two sentences, since such strong statements cannot be made based on only three events. For such strong conclusions, it is desirable to carry out hundreds of measurements of steps of each polarity or refer to articles where these results were obtained earlier.

Response: The comment is important for us! We reselected the examples and revised the section according to the reviewers' instructions. Thank you very much!

Comment: *Line 217-219. Comparing the morphology of the space leader shown in frames #1 and #3 and considering the frame interval is $1.11 \mu\text{s}$, the estimated backward velocity of the space leader is about $1.17 \times 10^5 \text{ m/s}$, and the estimated forward velocity of the descending negative leader is about $0.63 \times 10^5 \text{ m/s}$ during the phase.*

All velocities are average velocities and the authors would like to estimate the measurement

errors of all velocities (see lines 150-151).

Response: Thank you and the reviewer #1! We have revised the statement. Please see lines 190-193.

Comment: *Line 219-224. The estimated forward velocity of the descending leader is smaller than the estimated backward velocity of the space leader during the connection of the space leader to the primary leader channel, which is different from the positive leader steps shown in Figure 4. We infer that the smaller forward velocity of the descending primary leader may be the reason for the longer duration of the bi-directional development of the space leader in negative leader steps. Note that a common streamer zone could be seen in frame #2.*

It is advisable to delete this fragment, since you cannot make such strong statements based on just three events. For such strong conclusions, it is desirable to carry out hundreds of measurements of steps of each polarity. Moreover, these are very rough estimates in comparison with the classical works [Gorin et al. 1976, Les Renardières Group, 1977], in which measurements were made using continuous scanning of images and continuous measurement of velocities. The average speed significantly distorts the physics of the breakthrough phase, where the speed of the leaders' movement changes rapidly and is not constant.

Response: Under the direction of the comment, we realized the problem. We have removed the fragment in the revised manuscript. Thanks!

Comment: *Line 242-263.*

It is advisable to remove this section from the paper, since the authors wrote it so badly that it cannot be reviewed, since there are mistakes and misunderstandings in each sentence. Unfortunately, the review genre does not allow the reviewer to turn it into a lecture on plasma physics and long spark kinetics. The authors obtained an important experimental result, and this result does not weaken, the fact that the authors do not provide an adequate explanation, since at the moment no one has an adequate explanation of this phenomenon.

Response: We have deleted this section. We sincerely hope to receive more guidance from the reviewers in the follow-up!

Comment: *Line 265-266. The results presented above indicate that separated luminous structures in positive leader discharges could lead steps at high humidity, like that the space leader leads negative leader steps.*

It is reasonable for the authors to rewrite these lines, since the negative leader of a long spark in the discharge gaps of more than three meters always moves in steps, regardless of humidity.

Response: We have rewritten these lines! Please see lines 271-273. Thanks!

Comment: *Line 274-275. All these findings indicate that positive leaders could do steps in a similar way to negative leaders at high humidity.*

It is reasonable for the authors to rewrite these lines, since the negative leader of a long spark in the discharge gaps of more than three meters always moves in steps, regardless of humidity.

Response: We realized the problem! We have rewritten these lines. Please see lines 271-273. Thanks!

Comment: Line 275-276. One problem is solved, but more interesting questions are raised.

It is desirable to remove this sentence, since the problem cannot be solved on the basis of two observations. The study of the fine structure of the steps of a positive leader is just beginning.

Response: We have removed this sentence in the revised draft. Thank you for pointing the problem!

Comment: Line 281-282. All records presented in this paper were obtained at the Ultra-High-Voltage in this paper were obtained at the Ultra-High Voltage Laboratory in Hefei, China.

Response: Corrected! We are sorry for the mistake!

Comment: 280-281. Method.

It is desirable to move the "Method" section to the beginning of the paper. The information presented in this section needs to be supplemented:

- On all Figures it is desirable to give an oscillogram of voltage measurements.
- Where is the Marx generator located indoors or outdoors?
- How far from the tip of the HV-electrode were the high-speed cameras?
- What lenses were used for speed cameras?
- Have trigger delays been measured to accurately synchronize current, voltage, and exposure measurements on speed cameras?

Response: Thank you for your nice suggestions! However, depending on the requirements of the journal to which the manuscript is submitted, the "Method" section is placed at the end of the article. We have revised this section as per your guidance. We are sorry to say that in our experiments, the distance between the high-speed cameras and air gap was not recorded. We will improve this in future experiments. Thank you very much!

Thank the reviewer very much!

REVIEWERS' COMMENTS

Reviewer #1 (Remarks to the Author):

attached as pdf

This work has improved significantly. I appreciate the multiple examples of positive leaders that are now shown. I only have two significant comments. I believe the paper can be published without further review if these two small issues are fixed.

First, the authors are now referring to common streamer zones as “spindles”. This is not good, as it makes the paper more vague since the readers do not know what the authors mean by that term. The authors need to use already recognized terminology, or rigorously define terms they introduce. In this case I strongly suggest the authors use the term suggested by reviewer 3 in the previous review: “streamer-like common zone”. This has the advantage that it is clear how the authors are interpreting the observation and what it is that is being discussed.

Second, in lines L192 – 193 the authors give a bi-directional speed (as a range which is very good!). However, a “bi-directional speed” is very difficult to interpret. The authors should instead give the forward and backwards speeds separately (as ranges, four numbers total). This will help the authors next sentence where they claim that the forward speed was obviously greater than the backward speed.

Reviewer #2 (Remarks to the Author):

This is the third review of the manuscript by Huang et al. In my view since the first submission the scientific quality of the manuscript has improved: it now contains more useful details and now most of the issues that I and the other referees raised have been solved. Nevertheless I still have some comments that I think the authors may take into account to improve their manuscript.

1. In this latest version the discharge used to introduce the observations (figure 1) is different. In this case the “separate luminous structure” is not really well separated from the channel, which I think introduces confusion in the reader. Later, the authors show better examples of the phenomenon so I wonder if the more ambiguous figure 1 is the best case to illustrate the observations.

2. I still do not understand the emphasis put on differences between grayscale and color-coded images. In my view the grayscale one is not more authentic: both images represent a set of pixel values and how to present them is a presentation choice. Here it is clear that the color-

coded images offer higher contrast so I do not see why not using them everywhere. However, It would be nicer if the colorbar has easier to understand ticks (in steps of, say, 500 or 250 units).

3. In all pictures we see that the original resolution is 56x128 whereas later we find 128x56; why the reversal of dimensions?

Response to Reviewers

We would first like to express our gratitude to all the reviewers! They contributed significantly to this manuscript, and we have benefited a lot from their comments! We have revised the manuscript accordingly. Our responses are given in a point-by-point manner below.

Response to Reviewer #1

Comment: First, the authors are now referring to common streamer zones as “spindles”. This is not good, as it makes the paper more vague since the readers do not know what the authors mean by that term. The authors need to use already recognized terminology, or rigorously define terms they introduce. In this case I strongly suggest the authors use the term suggested by reviewer 3 in the previous review: “streamer-like common zone”. This has the advantage that it is clear how the authors are interpreting the observation and what it is that is being discussed.

Response: Thank you for pointing out the problem! We have replaced “spindles” with “streamer-like common zone” in the revised manuscript. Please see lines 32, 62, 133, 134, 140, 142, 214, 216, 237, and 281.

Comment: Second, in lines L192 – 193 the authors give a bi-directional speed (as a range which is very good!). However, a “bi-directional speed” is very difficult to interpret. The authors should instead give the forward and backwards speeds separately (as ranges, four numbers total). This will help the authors next sentence where they claim that the forward speed was obviously greater than the backward speed.

Response: Under the direction of the reviewer, the forward and backward speeds were estimated and presented separately in the revised manuscript. Please see lines 196-201. Thanks!

Thank the reviewer very much!

Response to Reviewer #2

Comment: In this latest version the discharge used to introduce the observations (figure 1) is different. In this case the “separate luminous structure” is not really well separated from the channel, which I think introduces confusion in the reader. Later, the authors show better examples of the phenomenon so I wonder if the more ambiguous figure 1 is the best case to illustrate the observations.

Response: The gap between the separate luminous structure and the abandoned tip of the branched leader is not evident. Your comments are right! This is really not a perfect example. But this example illustrates an important feature of the positive leader

steps led by separate luminous structures. The feature is that the channel junction is precedes the intense illumination and the sharp current rise. In addition, it could be deduced from physics that the separate luminous structure is not actually connected to the abandoned tip. (Assuming that the separate luminous structure is a part of leader channel quasi-continuously extending from the abandoned tip, this leader segment is charged positively. The selected tip of the branched leader would move away from this leader segment, due to the Coulomb force. The channel junction shown in frame #3 would not occur.) Therefore, we present the observations shown in figure 1.

To avoid possible confusion to the reader, we have added explanations in the revised manuscript. Please see lines 96-103. Thank you for pointing out the important issue!

Comment: I still do not understand the emphasis put on differences between grayscale and color-coded images. In my view the grayscale one is not more authentic: both images represent a set of pixel values and how to present them is a presentation choice. Here it is clear that the color-coded images offer higher contrast so I do not see why not using them everywhere. However, It would be nicer if the colorbar has easier to understand ticks (in steps of, say, 500 or 250 units).

Response: Thanks! Thank you for pointing out the problem! Only the color-coded images are presented in the revised manuscript, and the color bar had been adjusted. Thanks!

Comment: In all pictures we see that the original resolution is 56x128 whereas later we find 128x56; why the reversal of dimensions?

Response: We have made mistakes! The resolution of these frames shown in figure 1 is 56 (in width) × 128 (in height) pixels. We have corrected the problem in the revised manuscript. Please see the captions of Figure 1, Figure 3, Figure 4, and Figure 5. Thanks!

Thank the reviewer very much!